# The small and large ribosomal subunits depend on each other for stability and accumulation

Brian Gregory, Nusrat Rahman, Ananth Bommakanti, Md Shamsuzzaman, Mamata Thapa, Alana Lescure, Janice M Zengel, Lasse Lindahl

**The 1:1 balance between the numbers of large and small ribosomal subunits can be disturbed by mutations that inhibit the assembly of only one of the subunits. Here, we have investigated if the cell can counteract an imbalance of the number of the two subunits. We show that abrogating 60S assembly blocks 40S subunit accumulation. In contrast, cessation of the 40S pathways does not prevent 60S accumulation, but does, however, lead to fragmentation of the 25S rRNA in 60S subunits and formation of a 55S ribosomal particle derived from the 60S. We also present evidence suggesting that these events occur post assembly and discuss the possibility that the turnover of subunits is due to vulnerability of free subunits not paired with the other subunit to form 80S ribosomes.**

## Introduction

Ribosomes contain two different subunits, both of which are required for translation. The small subunit ("40S" in eukaryotes) decodes the genetic message and the large subunit ("60S" in eukaryotes) catalyzes peptide bond formation. The biogenesis of the eukaryotic 40S and 60S ribosomal subunits is a complex process that has been investigated most thoroughly in *Saccharomyces cerevisiae* (yeast). However, essential features are largely conserved from yeast to humans, even though significant complexity has been added in humans (Tafforeau et al, 2013; Tomecki et al, 2017). The process begins in the nucleolus, continues in the nucleoplasm, and is completed in the cytoplasm (Woolford & Baserga, 2013; Kressler et al, 2017; Bassler & Hurt, 2018). RNA polymerase I transcribes a precursor transcript that includes rRNA of the 40S subunit (18S) and two of the three rRNA components of the 60S subunit (5.8S and 25S), as well as transcribed spacers that are degraded during ribosome assembly. The third rRNA component of the 60S (5S) is polymerized separately by RNA polymerase III. During its transcription ("co-transcriptionally"), or soon after ("post-transcriptionally"), the long pre-rRNA is cleaved into two parts, each destined for one of the two ribosomal subunits (Osheim et al, 2004; Kos & Tollervey, 2010; Talkish et al, 2016), then is further processed into the mature rRNA moieties concurrently with assembly of r-proteins into pre-ribosomal large and small subunits. After this split, the pre-40S and pre-60S ribonucleoprotein particles assemble along separate pathways (Woolford & Baserga, 2013; Kressler et al, 2017). More than 250 protein and RNA factors are required to coordinate assembly, modify rRNA components, and control ribosomal morphopoiesis (Woolford & Baserga, 2013; de la Cruz et al, 2015; Turowski & Tollervey, 2015; Kressler et al, 2017). The incorporation of individual r-proteins into the nascent ribosomal subunits are interdependent with each other and with the action of ribosomal assembly and pre-rRNA cleavage factors (Woolford & Baserga, 2013; de la Cruz et al, 2015; Biedka et al, 2018), resulting in sequential association of the r-proteins. Although this hierarchy shows limited flexibility during changing growth conditions and mutational manipulation of the synthesis of assembly factors (Ohmayer et al, 2013; Talkish et al, 2016), r-proteins are referred to as "early," "middle," and "late" binding proteins depending on when in the assembly pathway they initially bind to the ribosomal precursor particles (Shajani et al, 2011; Gamalinda et al, 2014; de la Cruz et al, 2015; Fernandez-Pevida et al, 2016; Zhang et al, 2016).

Some ribosome biogenesis factors are required for production of both ribosomal subunits, whereas others are required for formation of only one of the two subunits (Woolford & Baserga, 2013; Klinge & Woolford, 2019). If a subunit-specific assembly factor or a ribosomal protein is depleted, the biogenesis of the corresponding subunit is abrogated, whereas the biogenesis of the other continues; this leads to assembly of unequal numbers of the two

Department of Biological Sciences, University of Maryland, Baltimore, MD, USA

Correspondence: lindahl@umbc.edu
Brian Gregory's present address is Duke University, Department of Biology, Durham, NC, USA
Nusrat Rahman's present address is Bentley University, Health Thought Leadership Network, Bentley University, Waltham, MA, USA
Ananth Bommakanti's present address is Trimasek Life Sciences Laboratory, Singapore
Md Shamsuzzaman's present address is Philips Research America, Cambridge, MA, USA
Mamata Thapa's present address is Medical Science Communications at Medtronic, Mansfield, MA, USA
Janice M Zengel's present address is Department of Biological Sciences, Carnegie-Mellon University, Pittsburgh, PA, USA

subunits and distortion of the normal 1:1 production of the two ribosomal subunits. Because mutations in human genes for r-proteins and subunit-specific biogenesis factors have been identified as a source of a disease class called ribosomopathies (Mattijssen et al, 2010; Narla & Ebert, 2010; Danilova & Gazda, 2015; Farley & Baserga, 2016; Bustelo & Dosil, 2018), it is important to know whether cells have mechanisms to rectify imbalanced production between the number of the two subunits. To gain insight into this largely unaddressed question, we used the yeast *S. cerevisiae* to investigate if specifically terminating the production of one subunit affects the accumulation of the other. Our results show that obstructing 60S subunit assembly inhibits accumulation of 40S subunits due to post-assembly turnover. On the other hand, inhibiting 40S assembly does not prevent 60S subunit accumulation, although, interestingly, it does result in fragmentation of the 25S rRNA and formation of 55S ribosomal particles derived from 60S subunits.

# Results

We interrupted assembly of the 40S or 60S ribosome subunits using strains in which the only gene for a given ribosomal protein or assembly factor is transcribed from the *GAL1/10* promoter. When these strains are shifted from galactose to glucose medium, the synthesis of the protein expressed from the *GAL* promoter stops, halting assembly of the subunit corresponding to the repressed protein gene. This allows us to address the question of whether inhibition of assembly of one subunit affects accumulation of the

other. We refer to these strains as $P_{gal}$-xx, where xx is the name of the protein expressed from the *GAL* promoter. Because the shift from galactose to glucose medium superimposes a nutritional shift-up on the repression of ribosomal genes, we performed control experiments with the BY4741 strain in which all ribosomal genes are expressed from their natural chromosomal genes.

Even though the shift from galactose to glucose prevents further assembly of one of the subunits, the culture continues to grow (as determined by the culture OD$^{600}$), albeit at a steadily declining rate, because the ribosomes made before the shift remain active in protein synthesis (Figs 1A and S1). Because cessation of r-protein synthesis arrests the cells in early G1 phase and results in morphological changes (Thapa et al, 2013; Shamsuzzaman et al, 2017), we wanted to determine if the cells are still viable. We, therefore, compared colony-forming units with the number of cells visible under the microscope after repressing uL22 synthesis (Fig 1B). By these criteria, approximately 80% of the cells form colonies during growth in galactose medium as well as after the shift to glucose. Thus, the interference with ribosome assembly does not affect cell viability.

### Abrogating assembly of the 60S subunit inhibits accumulation of 40S subunit

We first disrupted the 60S assembly pathway at different points by selectively repressing genes for early (uL4), middle (uL22), or late binding proteins (uL18 and eL43), all of which bind to 60S precursor particles (pre-60S) in the nucleolus or nucleus (Gamalinda et al,

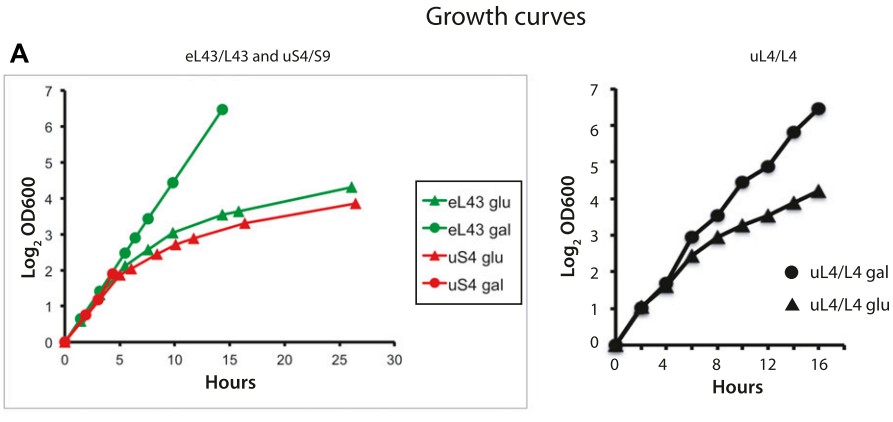

**Figure 1. Growth curves and cell viability.**
**(A)** $P_{gal}$-S4, $P_{gal}$-eL43, and $P_{gal}$-uL4 were grown in YEP-galactose medium. At time 0, one part of each culture was shifted to glucose medium (YPD), and the OD$_{600}$ of the unshifted and shifted cultures was recorded at the indicated times. **(B)** Pgal-uL22 and the control strain BY4741 were shifted from galactose to glucose medium. Aliquots of the unshifted and shifted cultures were removed, and the total number of cells was counted under a microscope and viable cells (colony-forming units) were determined by plating on YEP-galactose medium.

2014; de la Cruz et al, 2015). We also depleted 60S assembly factors Rpf2 or Rrs1; these proteins mediate the binding of r-proteins uL5 and uL18 as well as 5S rRNA to the nascent 60S subunit in the nucleus (Zhang et al, 2007).

Sucrose gradient $A^{260}$ profiles of lysates from galactose cultures were essentially the same for all mutant strains as for BY4741 (compare Fig 2A with Fig 2C, E, I, and K and with Fig S2A and C). However, 8–16 h after the shift to glucose, sucrose gradients of all mutant extracts were very different from the glucose-shifted control BY4741 (Fig 2, compare panel B with panels D, G, H, J, L

and with Fig S2B and D): The 60S peak in mutant extracts was markedly reduced relative to other ribosomal peaks, as expected because the 60S assembly was blocked. Furthermore, all ribosomal peaks became smaller relative to the $A^{260}$ material remaining at the top of the gradient, showing that the ribosome content was diluted during the continuing growth after ribosome accumulation was inhibited (compare Fig S2M and N). Importantly, the same result was seen irrespective of which 60S r-protein or assembly factor gene was repressed, showing that the effect does not depend on which step in the 60S assembly pathway was disrupted.

## No gene repression (BY4741)

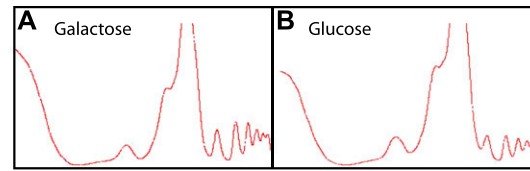

## Gene repressed for

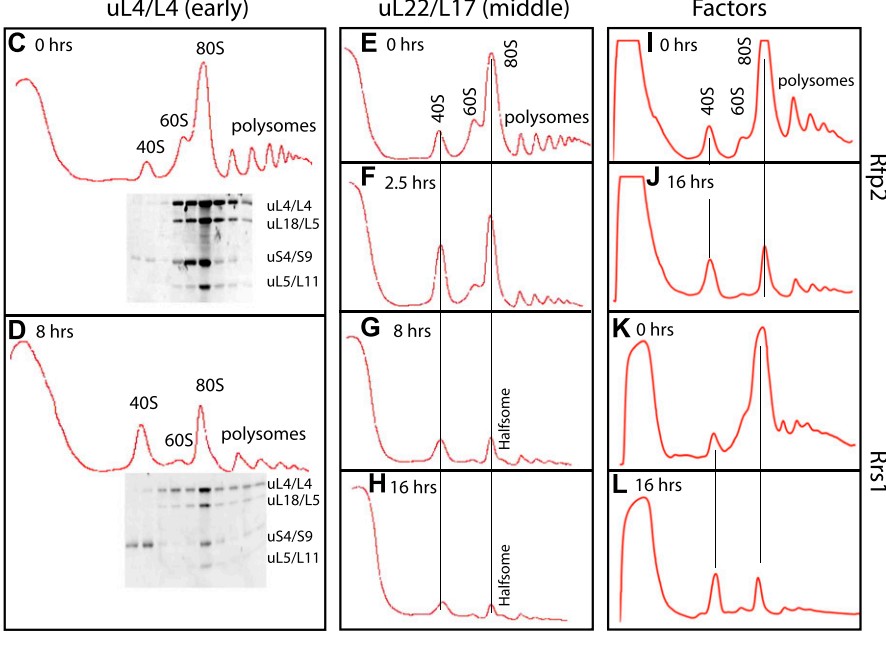

## Quantification of 40S/total ribosomes

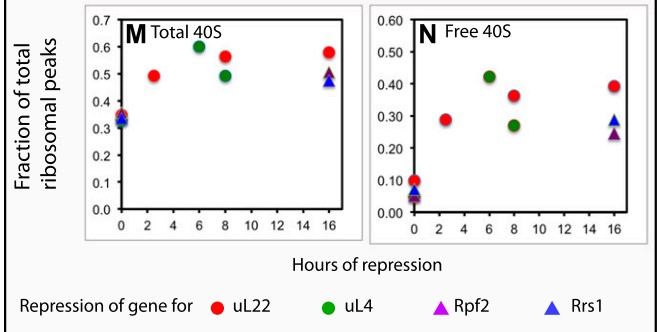

**Figure 2. Sucrose gradient analysis of ribosomes after repressing the genes for 60S r-proteins uL4 and uL22 and 60S assembly factors Rpf2 and Rrs1.**
Cultures were grown in YEP-galactose medium and shifted to YPD glucose medium for the indicated lengths of time before harvest. **(A, B)** The control strain BY4741 in galactose and glucose medium, respectively. **(C, D)** $P_{gal}$-uL4 at 0 and 8 h, respectively. **(E–H)** $P_{gal}$-uL22 at 0, 2.5, 8, and 16 h, respectively. **(I, J)** $P_{gal}$-Rpf2 at 0 and 16 h, respectively. **(K, L)** $P_{gal}$-Rrs1 at 0 and 16 h, respectively. For C and D, equal volumes of fractions in the ribosome portion of the gradient were analyzed for r-proteins uS4, uL4, uL5, and uL18 by Western blot. **(M, N)** Quantification of 40S subunits. The area under each ribosomal peak was measured using ImageJ and normalized to the area under all ribosomal peaks. **(M)** The fraction of ribosomal mass in 40S subunits was calculated as the fraction found in the free 40S peak plus 1/3 of the fraction in 80S and polysomes. **(N)** The fraction of ribosomal mass in free 40S subunits. Red and green circles refer to repression of uL22 and uL4 synthesis, respectively. Purple and blue triangles refer to repression of Rpf2 and Rrs1 synthesis, respectively.

The interference with 60S assembly also causes an increase in the 40S peak (compare Fig 2 panels C versus D, E versus F–H, I versus J, K versus L and Fig S2 panels A versus B and C versus D), raising the possibility that this peak includes uncommon precursor particles or degradation derivatives of 60S subunits. Therefore, we determined the distribution of several r-proteins in sucrose gradients loaded with extracts harvested before and 8 h after repression of uL4 (Fig 2C and D). In either case, the 40S peak contained the 40S protein uS4, but none of the 60S proteins tested (uL4, uL5, and uL18). We conclude that no 60S-related particles co-fractionate with 40S subunits, indicating that the 40S peak contains only free 40S subunits. Thus, the increased 40S peak suggests a buildup of 40S subunits that cannot combine with 60S to form 80S because 60S assembly was blocked.

If the accumulation of 40S subunits continues after abolishing 60S assembly, the excess of 40S subunits should be progressively larger with time. To test this expectation, we quantified the ribosomal peaks in sucrose gradients loaded with lysates of cells harvested at different times after repressing uL22 synthesis (Fig 2E–H). This shows that the ratio of total 40S to total ribosome mass did in fact increase by 50–60% during the first couple of hours after repressing uL22 synthesis, but then became virtually constant (Fig 2M). Because the 60S accumulation is inhibited, the excess 40S cannot combine with 60S subunits and accumulate as free 40S, but the amount of free 40S relative total ribosomal mass also became constant after a few hours (Fig 2N). Importantly, quantification of ribosomal peaks after repressing uL4, Rpf2, and Rrs1 for the indicated times fall on the same curve as the repression of uL22, confirming that the results do not depend on how the 60S assembly is disrupted.

Finally, we note that sucrose gradients loaded with lysates from cells harvested after repression of 60S r-protein genes have been described in previous publications addressing the mechanisms of ribosome assembly in yeast and humans, but the authors focused on the mechanism of ribosome assembly, not on the accumulation of the ribosomal subunits (e.g., Robledo et al (2008); Gamalinda et al (2012); Jakovljevic et al (2012); Wan et al (2015)).

## Quantification of ribosomal components confirms that 40S accumulation stops after inhibiting of 60S assembly

The plateauing of the imbalance between the ribosomal subunits suggests that blocking 60S assembly halts the accumulation of not only 60S subunits but also 40S subunits. To test this notion, we measured the specific concentration of several r-proteins (r-protein$_i$/total protein, hereafter called S-conc). If the accumulation of both subunits is blocked, S-conc of both 40S and 60S r-proteins should decrease after cessation of 60S assembly because the ribosomes made before the inhibition of subunit formation continue to make total cellular protein after the insult to subunit assembly. Accordingly, we determined the S-conc of r-proteins from both subunits by Western blot analysis. Because disrupting ribosome biogenesis causes pleiotropic changes in the expression of numerous genes (Shamsuzzaman M, Gregory B, Bruno V, and Lindahl L, in preparation), we could not identify a protein suitable for Western loading control. We, therefore, used $A^{280}$ units to normalize the intensity of the Western bands. We acknowledge that the amount of protein per $A^{280}$ unit might vary because of the changing protein/RNA ratio during depletion of r-proteins, as RNA contributes to $A^{280}$. However, probing the same blot with antisera against uS4, uL4, uL5, and uL18 and comparing the bands for the different r-proteins in the same samples eliminates this issue and directly shows if proteins covary. Constant numbers of $A^{280}$ units of lysates from three independent cultures harvested before and at different times after the shift to glucose medium were loaded in slots on the same gel and transferred to a membrane after electrophoresis. The blot was then probed with a cocktail of antisera against uS4, uL4, uL5, and uL18. Because the antisera had different avidities, the intensity of the bands was different for each protein. In some experiments, we, therefore, first probed with the weakest antisera, amplified the image electronically, and then probed with the rest of the antisera without stripping.

As seen in Figs 3A, E, and S3A, the S-conc of the r-proteins does not change significantly after the shift to glucose of the control strain (BY4741), and the ratio between the different r-proteins does not change. After repressing the 60S r-protein genes encoding uL4, uL11, or eL40, the S-conc of 60S r-proteins uL4, uL5, and uL18 all decrease (Figs 3B–D and S3B–D) in parallel (Fig 3F–H; see below about uL18 after cessation of uL4 synthesis). This was anticipated because blocking the supply of any essential 60S protein prevents full assembly and thus prevents incorporation of all 60S proteins into a stable ribosomal complex. Significantly, the uS4 S-conc also decreases (Figs 3B–D and S3B–D) and covaries with S-conc of the 60S proteins (Fig 3F–H). This result is consistent with our model that 40S accumulation is inhibited by the prevention of 60S assembly. Interestingly, uL18 decreases slower in the uL4-repressed cells (Fig 3B and F) than in the eL40-repressed cells (Fig 3D and H). This may be related to the fact that uL18 is incorporated into an extra-ribosomal particle with 5S rRNA and uL5 before it is transferred to the pre-60S (Zhang et al, 2007, see also the Discussion section). Finally, we point out the decline of the S-conc of uL4 follows the dilution curve calculated from the growth curve (Fig 3B), showing that the decline in the S-conc of uL4 can be explained by dilution and does not require invoking active degradation, for example, by proteasomes.

To further test the model that inhibition of 60S assembly prevents 40S accumulation, we compared the abundance of 18S and 25S rRNA in the cells. By the same logic applied to the r-proteins above, our model predicts that the ratio of rRNA remains relatively constant after blocking 60S assembly. Total RNA purified by affinity chromatography after repression of the uL4 or uL18 gene was fractionated by agarose gel electrophoresis and stained with methylene blue. Consistent with our prediction, no obvious change in the 18S/25S ratio is seen after repressing the 60S protein genes (Fig 3I). We further analyzed the 18S/25S ratio by gel electrophoresis of RNA purified by hot phenol extraction and quantification of ethidium bromide–stained bands. As expected, the ratio between 18S and 25S rRNA did not change after shifting the control strain (BY4741) to glucose medium (Figs 3J and S4A). However, after repression of 60S r-protein genes, the 18S/25S ratio increased 20–50%, then either stabilized (after repressing uL4 or uL18 synthesis) or declined back to 1:1 (after repressing the eL40 gene) (Figs 3J and S4B–D). This is in agreement with the sucrose gradient experiments that showed an initial ~50% increase relative to total ribosomal mass, but then no further increase.

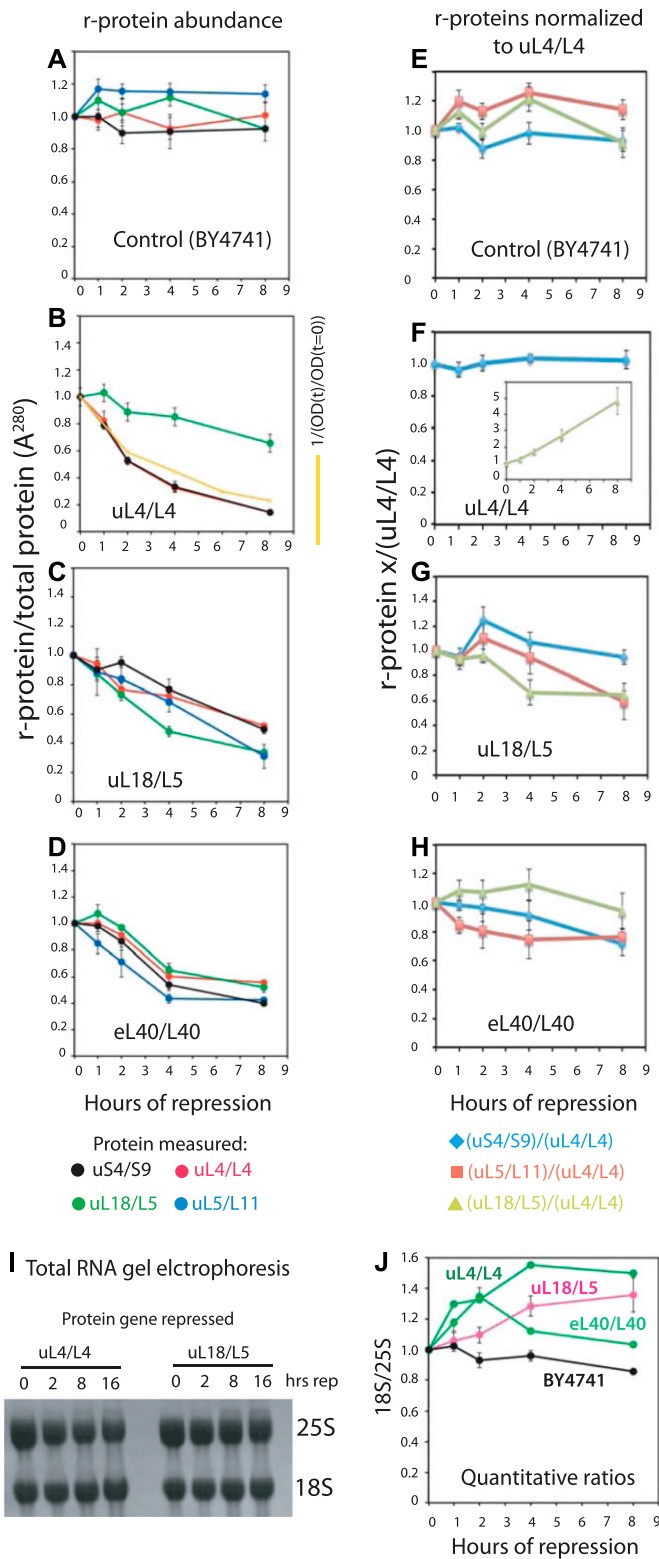

**Figure 3. Quantification of ribosomal components from both subunits covary after repression of genes for 60S r-proteins.**

Cultures of BY4741, P$_{gal}$-uL4, P$_{gal}$-uL18, and P$_{gal}$-eL40 were grown in YEP-galactose and shifted to YPD. **(A–D)** Aliquots were harvested at 0, 1, 2, 4, and 8 h and constant A$^{280}$ units from each sample were analyzed by Western blots probed with antisera for uS4, uL4, uL5, and uL18. The intensities of the Western bands were quantified

## Abrogating assembly of the 40S subunit does not inhibit 60S accumulation

We next tested the effect of preventing 40S subunit assembly by repressing the synthesis of early (uS4 and uS11), middle (uS7), and late (uS10 and eS31) 40S assembly r-proteins, which all bind to pre-40S in the nucle(o)lus (de la Cruz et al, 2015; Fernandez-Pevida et al, 2016). We also analyzed 40S assembly factor Rrp7, which works in the early steps of the assembly (Perez-Fernandez et al, 2007). We observed the same pattern regardless of how 40S assembly was inhibited, namely, the 40S peak is reduced and the 60S peak increases relative to the 80S peak (Figs 4A–K and S2E–L). Although this increase is evident, the overlap between the 60S and 80S prevents an accurate quantification of these peaks. Furthermore, the ratio between the ribosomal peaks and the A$^{260}$ material at the top of the gradient decreases less after repression of 40S assembly than it did after repressing 60S assembly (compare Fig S2 panels M, N with panels O, P). These results suggest that 60S assembly and accumulation continue during interruption of 40S assembly. To test this, we constructed a strain in which uS17 is under the control of the galactose promoter and GFP-tagged uL23 is expressed from an estradiol-inducible promoter. This strain was grown in galactose and shifted to glucose for 8 h. Synthesis of uL23-GFP was then induced by addition of β-estradiol. As seen in Fig 4L, M, uL23-GFP is incorporated into mature 60S subunits that combine with 40S to form 80S, showing that functional 60S subunits are formed during inhibition of uS11 synthesis. Although 60S accumulation continues in the absence of 40S production, we were intrigued to observe the buildup of a new peak, sedimenting slightly slower than 60S. We refer to this new peak as "55S" (Fig 4 panels B, C, F–I, and K, and Fig S2F, H, J, L, and P) and will describe it in more detail below.

As with our analysis of the effect of repressed 60S assembly on r-protein accumulation (Fig 3), we also quantified the specific concentrations of r-proteins after cessation of 40S assembly. The response was strikingly different from the abrogation of 60S assembly: after abolition of the synthesis of uS4, uS10, or uS11, only the uS4 S-conc decreases, whereas the S-conc of 60S proteins uL4, uL5, and uL18 changed little (Figs 5A–C and S3E–G) and the S-conc of the 60S proteins covary (Fig 5D–F). This supports our conclusion that abrogation of 40S assembly does not inhibit 60S accumulation. Furthermore, the S-conc of uS4 essentially follows the dilution curve calculated from the growth curve (Fig 5A), indicating that reduced amount of uS4 can be explained by growth-related dilution of ribosomes made before the shift to glucose medium. We noted above a similar correlation for the repression of uL4 synthesis.

The 18S/25S ratio also changed in accordance with the model that abolition of 40S assembly does not affect 60S accumulation.

using ImageJ. The yellow curve in (panel B) indicates the calculated dilution curve after repressing uL4 synthesis. **(E–H)** Data from A–D were normalized to the values for uL4. **(I, J)** Ratio between 18S and 25S rRNA. Total RNA was purified using RiboPure (I) or hot phenol extraction (J). Equal A$^{260}$ units of each sample (0.06 corresponding to ~3 μg) were applied to agarose gels (I). Bands were blotted to a membrane and stained with methylene blue. **(J)** Gel was stained with ethidium bromide, photographed on a gel imaging system, and quantified with ImageJ. Images of the Western blots are shown in Fig S3. Fig S4 shows images of the ethidium bromide stained gels.

Source data are available for this figure.

## Gene repressed for

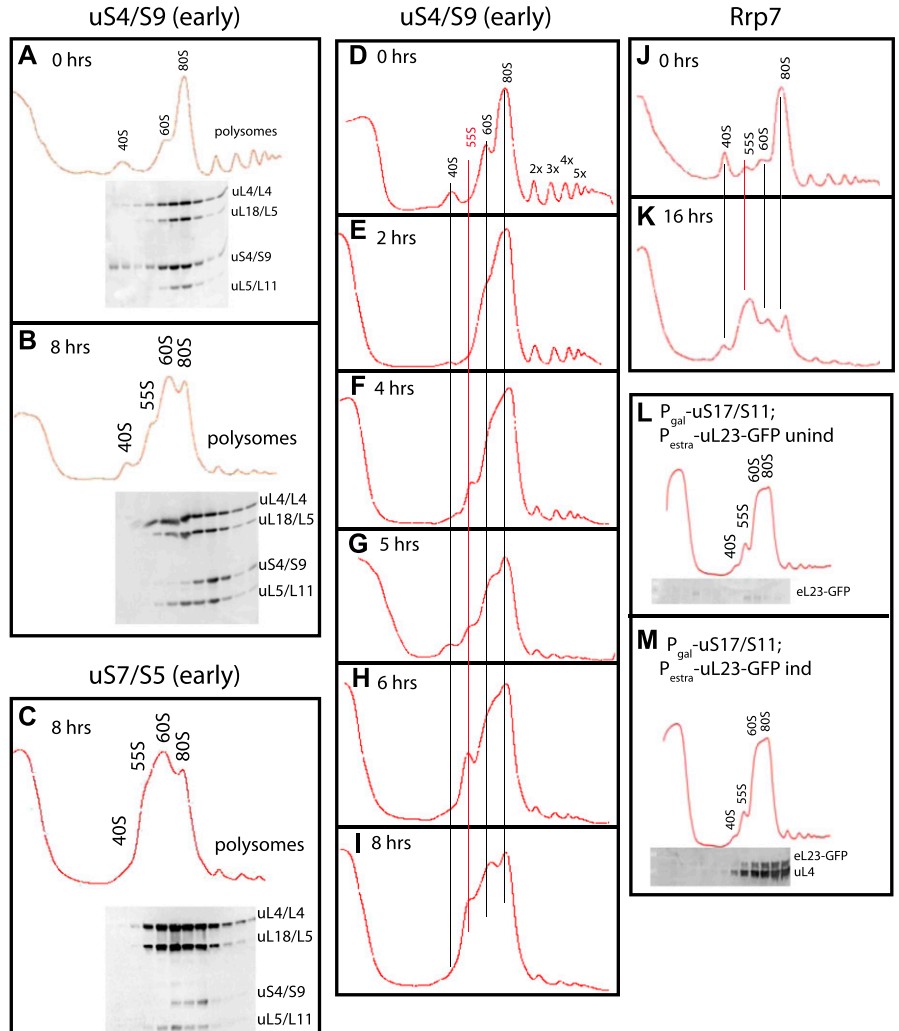

**Figure 4. Sucrose gradient analysis of ribosomes after repressing the genes for the 40S r-proteins uS4 and uS7 and 40S assembly factor Rrp7.**
Cultures in YEP-galactose medium were shifted to glucose medium for the indicated lengths of time before harvest. **(A, B)** $P_{gal}$-uS4 at 0 and 8 h, respectively. **(C)** $P_{gal}$-uS7 at 8 h **(D–I)** $P_{gal}$-uS4 at 0, 2, 4, 5, 6, and 8 h, respectively. **(J, K)** $P_{gal}$-Rrp7 at 0 and 16 h, respectively. For A–C, equal volumes of fractions in the ribosome portion of the gradient were analyzed for r-proteins uS4, uL4, uL5, and uL18 by Western blot. **(L, M)** uL23-GFP synthesized during repression of the uS17 gene is incorporated into 80S ribosomes. $P_{gal}$-uS17 harboring an estradiol-inducible uL23-GFP gene was grown in synthetic galactose medium and shifted to synthetic glucose medium. 8 h after the shift, uL23-GFP synthesis was induced for 1 h. The indicated fractions were probed by Western analysis for GFP and uL4 proteins.

Electrophoresis and methylene blue staining of total RNA after repressing uS4 synthesis showed a clear reduction of 18S rRNA relative to 25S rRNA (Fig 5G). This conclusion was borne out by quantification of the 18S/25S ratio on ethidium bromide stained gels. The ratio declined four- to fivefold after repression of the genes for uS4, uS10, or uS11 (Figs 5H and S4E–G). The subtle variances in the kinetics of the 18S/25S ratio after the abolition of different proteins from the same subunit are most likely due to differences in the rate of turnover of abortive assembly intermediates depending on which step of the assembly is affected by the cessation of individual r-proteins.

### 40S assembly continues after blocking 60S assembly

The plateauing of the number of 40S relative to total ribosomes (Fig 2M and N) implies that either 40S assembly stops after a few hours, or it is offset by degradation of 40S subunits. To distinguish between these possibilities, we examined the transcription and processing of

rRNA after the cessation of 60S assembly. First, we quantified the cell content of internal transcribed spacers (ITS1 and ITS2) relative to total RNA. These parts of the primary rRNA transcript are degraded during subunit manufacturing. The abundance of transcribed spacer sequences will decline rapidly if no new transcripts are made, because the lifetime of rRNA precursor molecules harboring the transcribed spacers is only 10–20 min (see e.g., Deshmukh et al (1993); Kressler et al (1997) and Fig 6G). We prepared total RNA at different times after shifting Pgal-uL4, Pgal-uS4, and the control strain BY4741 from galactose to glucose and then loaded a constant number of $A^{260}$ units onto a membrane in a slot pattern ("slot blot," Fig 6A). We quantified the abundance of ITS1 and ITS2 segments by probing the blots with $^{32}$P-end–labeled oligonucleotides O1663 and O1660 (complementary to sequences in ITS1 or ITS2, respectively [Fig 6H]).

Fig 6B and C shows that, after the shift from galactose to glucose, the abundance of molecules containing ITS1 or ITS2 relative to total RNA increases 2.5–3 fold after the shift of the control strain as well as $P_{gal}$-uL4 and $P_{gal}$-uS4, although the ratio is slightly lower when

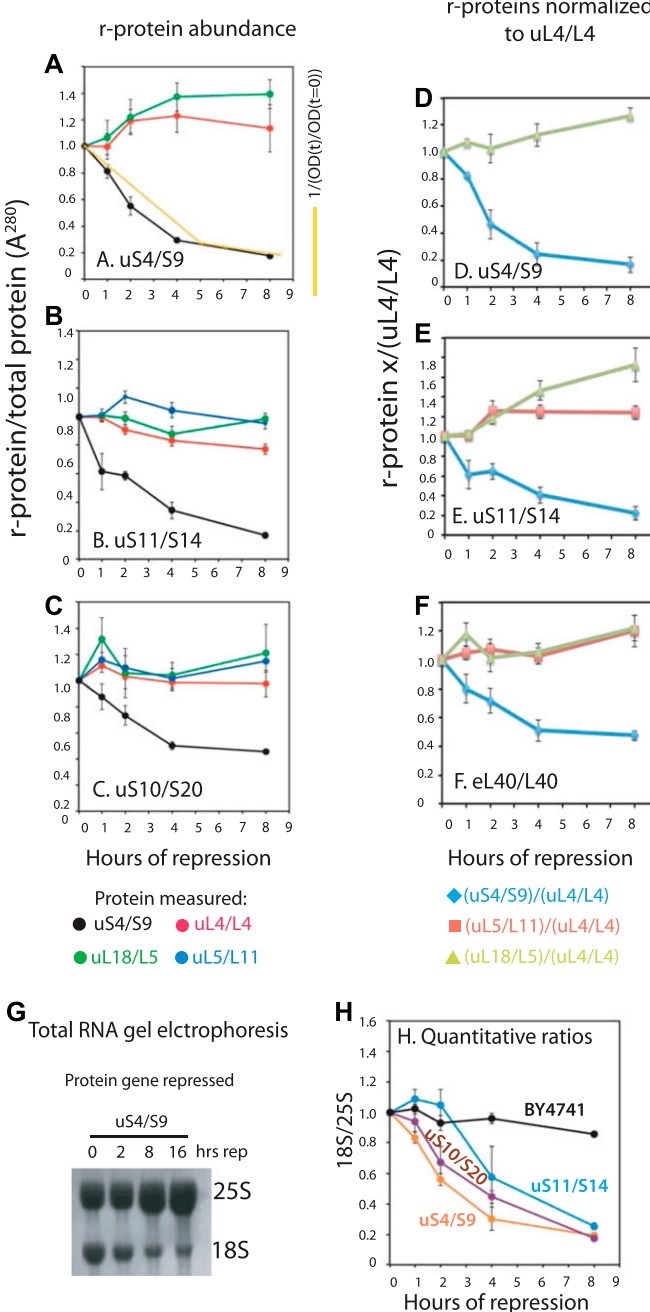

**Figure 5. Quantification of ribosomal components from both subunits do not covary after repression of genes for 40S r-proteins.**

Cultures of P$_{gal}$-uS4, P$_{gal}$-uS11, and P$_{gal}$-uS10 were grown in YEP-galactose and shifted to YPD. The control experiment with BY4741 is shown in Fig 3A and E. **(A–C)** Aliquots were harvested at 0, 1, 2, 4, and 8 h and equal A$^{280}$ units from each sample were analyzed by Western blots probed with antisera for uS4, uL4, uL5, and uL18. The intensities of the Western bands were quantified using ImageJ. The yellow curve in panel A indicates the dilution curve calculated from the growth curve after repressing uS4 synthesis. **(D–F)** Data from A–C were normalized to the values for uL4. **(G, H)** Ratio between 18S and 25S rRNA. Total RNA was purified using RiboPure (G) or hot phenol extraction (H). Equal A$^{260}$ units of each sample (0.06 corresponding to ~3 μg) were applied to agarose gels. **(G)** Bands were blotted to a membrane and stained with methylene blue. **(H)** Gel was stained with ethidium bromide, photographed on a gel imaging system, and quantified with ImageJ. Images of the Western blots are shown in Fig S3. Fig S4 shows images of the ethidium bromide stained gels.

the assembly of the subunits is blocked than during uninhibited ribosome formation. This increase is likely caused by the increased rRNA synthesis relative to total cellular RNA (d(rRNA)/(total RNA)) triggered by the switch of carbon source ("shift-up"), but a change in the rate of degradation of the transcribed spacers may also contribute to the changing ratio. Assuming that the rate of ITS1 and ITS2 degradation is similar whether or not ribosomal subunit assembly is inhibited, the measurements of the internal transcribed sequences thus shows that the transcription of rRNA relative to total cellular RNA is not significantly decreased by the repression of r-protein synthesis.

The presence of ITS1 sequences after repressing uL4 synthesis is also evident from primer extension analysis. The oligonucleotide O1345, which is complementary to a segment of the 5.8S gene (Fig 6H), was hybridized to equal amounts of total RNA harvested at the indicated times after repressing uL4. The results show that 5′ RNA ends produced by cleavages at the A2 and A3 sites in ITS1 are found in RNA harvested 8 or 18 h after repressing uL4 synthesis; their amounts are comparable with the amounts in RNA prepared before the shift (Fig 6D), even though the 5′ ends of the short and long form of 5.8S rRNA decline when 60S assembly is blocked because its maturation is blocked by the lack of new uL4 synthesis. Because the transcribed spacers are eliminated from newly transcribed rRNA within 20 min (Fig 6G, see also below), the analysis of ITS1 and ITS2 therefore shows that rRNA transcription continues for at least 18 h after inhibiting subunit assembly.

Run-on experiments also showed that rRNA transcription relative to the total cell content of RNA changes little after repression of uL4 synthesis. Cell extracts were prepared before and at 6 and 15 h after repressing uL4 synthesis under conditions that preserve transcribing RNA polymerase molecules on their template. Equal A$^{260}$ units of extracts from each sample were then incubated in the presence of radioactive UTP under conditions that allow RNA polymerase I to complete its current round of transcription, but not to initiate new rounds of transcription. The radioactive run-on products were purified and hybridized to a membrane loaded with denatured pDK16 plasmid DNA containing the full polymerase I 35S rRNA transcription unit (Lindahl et al, 1994). As seen in Fig 6E, the intensities of the rRNA run-on transcripts were very similar to the control strain both before and after the shift, indicating that, even 15 h after blocking 60S assembly, rRNA transcription relative to the total cell RNA content does not change.

To estimate the absolute rRNA transcription rate from these data, we measured the amount of A$^{260}$ material in lysates relative to the OD$_{600}$ of the culture after repressing uL22 synthesis. Fixed amounts of OD$_{600}$ units were harvested, and total RNA was prepared by the hot phenol method. Fig 6F shows that the A$^{260}$ yield per OD$_{600}$ unit increased after the shift of the control strain, as expected because of the nutritional shift from galactose to glucose. However, the ratio changed little after repressing the uL22 gene, suggesting that the abrogation of ribosome formation is offset by a shift-up effect on the synthesis of other forms of RNA. Potentially, a pool of rRNA degradation fragments could also contribute to the constant yield of A$^{260}$ material per OD$_{600}$ of culture harvested. Whatever the reason, the rate of rRNA synthesis is constant relative to total RNA and must, therefore, change in proportion to the culture growth rate, as determined by the OD$_{600}$ of the culture and, thus, gradually declines about twofold during 16 h of repression of uL4 synthesis (Fig 1A).

It is known that rRNA is unstable if it is not assembled into subunits (Kressler et al, 1997; Billy et al, 2000; Emery et al, 2004;

## Measurement of ITS1 and ITS2 concentration

**A**

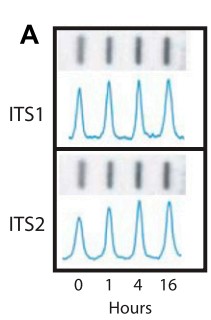

ITS1

ITS2

0  1  4  16
Hours

**B**

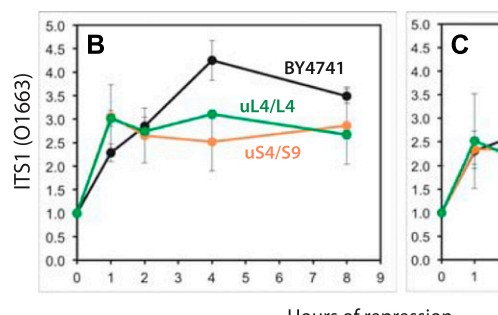

ITS1 (O1663)

BY4741
uL4/L4
uS4/S9

Hours of repression

**C**

ITS2 (O1660)

BY4741
uL4/L4
uS4/S9

Hours of repression

**D** Primer extension

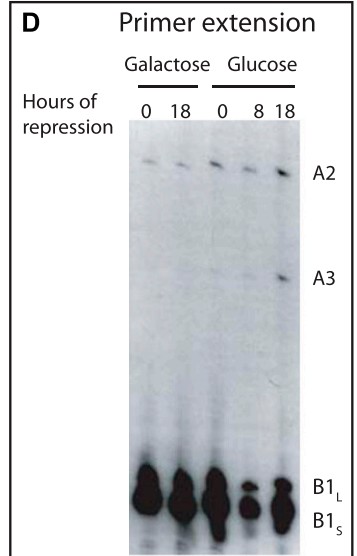

Galactose | Glucose
Hours of repression
0  18  0  8  18

A2
A3
B1$_L$
B1$_S$

**E** rRNA run-on

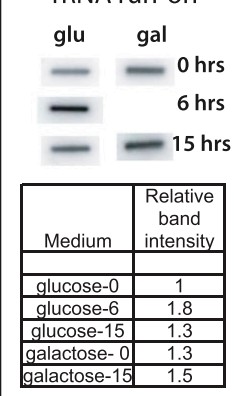

glu    gal
0 hrs
6 hrs
15 hrs

| Medium | Relative band intensity |
|---|---|
| glucose-0 | 1 |
| glucose-6 | 1.8 |
| glucose-15 | 1.3 |
| galactose- 0 | 1.3 |
| galactose-15 | 1.5 |

**F** RNA per cell

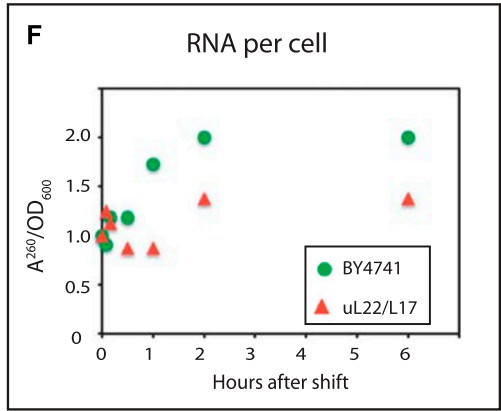

$A^{260}/OD_{600}$

BY4741
uL22/L17

Hours after shift

**G** rRNA processing

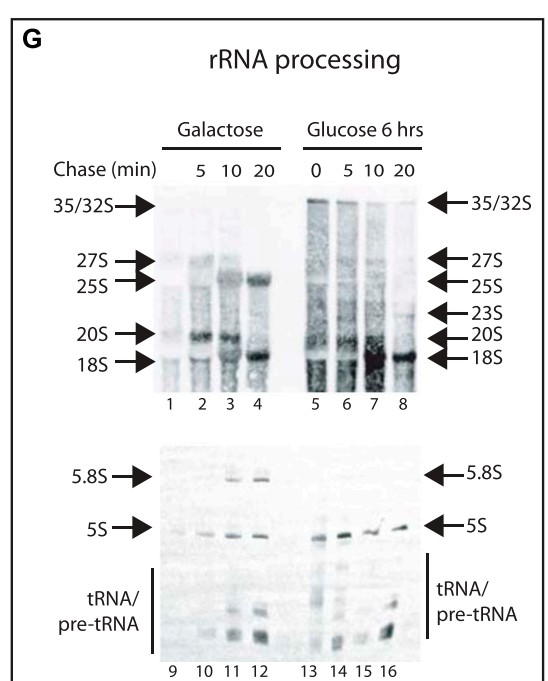

Galactose | Glucose 6 hrs
Chase (min)    5  10  20    0  5  10  20

35/32S → ← 35/32S
27S → ← 27S
25S → ← 25S
← 23S
20S → ← 20S
18S → ← 18S
1  2  3  4    5  6  7  8

5.8S → ← 5.8S
5S → ← 5S

tRNA/pre-tRNA    tRNA/pre-tRNA
9  10  11  12    13  14  15  16

**H** Map of PolI transcript

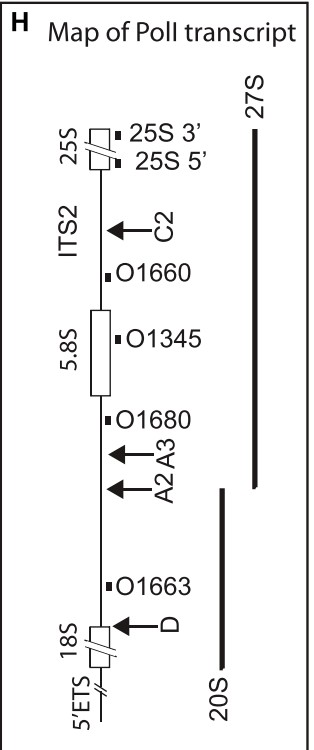

27S
25S
25S 3'
25S 5'
ITS2
C2
O1660
5.8S
O1345
O1680
A2 A3
18S
O1663
D
5'ETS
20S

Horsey et al, 2004; Horn et al, 2011; Woolford & Baserga, 2013; Talkish et al, 2016). Repressing uL4 synthesis is, therefore, expected to abort formation of mature 25S rRNA. However, the synthesis of 40S proteins is not inhibited and assembly of the 40S subunit and formation of 18S rRNA could, therefore, continue because uL4 is not part of the 40S subunit. To determine if the final 18S product is made in the absence of 60S assembly, we pulse-labeled cells with [3]H-uracil for 2 min, at which time a large excess of nonradioactive uracil was added. Total RNA was prepared before and 5, 10, and 20 min after addition of an excess of nonradioactive uracil. Equal amounts of radioactivity from each preparation were fractionated by agarose and acrylamide gel electrophoresis, and the rRNA bands were visualized by fluorography (Fig 6G). Before repression of uL4 synthesis, we observed the expected pattern, that is, over a 10–20-min period, the radioactive rRNA was first seen in the 35S primary transcript, then moved to 27S and 20S processing intermediates, and finally to mature 25S, 18S rRNA, and 5.8S rRNA (Fig 6G, lanes 1–4 and 9–12). 6 h after repression of uL4 synthesis, no mature 25S rRNA or 5.8S rRNA was formed (lanes 4–8 and lanes 13–16), but 18S was still matured to its final size (compare lanes 5–8 with lanes 1–4). We note that the 35S and 32S are increased in response to the cessation of 60S assembly, indicating that the initial cleavages of the pre-rRNA are delayed. This is likely caused by a change in the balance between co-transcriptional and post-transcriptional rRNA processing (Axt et al, 2014; Talkish et al, 2016) and does not necessarily affect the throughput of pre-rRNA to the subunit whose assembly is uninhibited. In fact, because the final amount 18S rRNA is very similar before and after inhibiting uL4 synthesis, our results show that the efficiency of 18S rRNA formation from the precursor transcript is not affected significantly by repressing uL4, as was previously shown after repressing the synthesis of uL1/L1 (Deshmukh et al, 1993). The sustained processing of 18S rRNA indicates that 40S subunits are assembled in the absence of uL4 synthesis, even though we cannot exclude that modification of rRNA and/or r-proteins is incomplete. We, therefore, propose that even though 40S subunits do not accumulate during inhibition of 60S assembly, the formation of 40S continues. Thus, the fading 40S accumulation a few hours after blocking synthesis of 60S r-proteins must be due to the turnover of (nearly) mature 40S subunits.

Interestingly, 5S rRNA processing is not affected by the disruption of 60S assembly (Fig 6G, compare lanes 9–12 with lanes 13–16). We have previously observed that inactivation of the endonuclease RNase MRP stops processing of both the 18S and 25S/5.8S segments of the 35S precursor rRNA, thereby blocking maturation of 18S, 5.8S, and 25S rRNA, but does not affect 5S rRNA accumulation (Lindahl et al, 2009). Possibly, 5S rRNA maturation is independent of 60S assembly because the 5S rRNA is initially incorporated into an extra-ribosomal particle with r-proteins uL5 and uL18 and assembly factors Rpf2 and Rrs1 (Zhang et al, 2007).

### Constraining 60S assembly inhibits nuclear export of both pre-40S and pre-60S particles, but 40S assembly inhibition affects only pre-40S export

Ribosomal precursor particles are exported from the nucleus to the cytoplasm once they become competent to bind nuclear export factors (Fischer et al, 2015; Malyutin et al, 2017). If the assembly is blocked, r-proteins accumulate in the nucleus (Hurt et al, 1999; Milkereit et al, 2003). To determine if the disruption of 40S or 60S assembly affects nuclear export of subunit precursors, we introduced low copy plasmids carrying uS5-GFP or uL23-GFP fusion genes expressed from the native promoters into the $P_{gal}$-eS31 and $P_{gal}$-eL43 strains and performed confocal microscopy of the resulting strains 16 h after the shift to glucose medium. Fig 7 shows that repression of the eS31 gene results in nuclear accumulation of uS5-GFP, but not of uL23-GFP, confirming that preventing assembly of the pre-40S precursor to an export-competent stage specifically traps 40S protein in the nucleus. As also expected, uL23 also accumulated in the nucleus after blocking pre-60S assembly by repressing the eL43 gene. It was, however, surprising that repression of the eL43 gene also caused nuclear accumulation of uS5. Thus, abolishing 60S assembly inhibits export of both subunits, although we cannot decipher if this is a direct or indirect result of inhibiting 60S assembly.

### Vacuole expansion

In agreement with previous observations (Bernstein et al, 2007), extremely large vacuoles develop when assembly of either subunit is constrained (Fig 7). Because the cell viability does not decline significantly over 16 h of inhibition of ribosomal subunit assembly

**Figure 6. Transcription and maturation of rRNA after repression of 40S and 60S r-protein synthesis.**
**(A–C, F)** The control strain BY4741 and strains in which the indicated proteins were expressed from the galactose promoter were grown in YEP-galactose (A–C, F) or synthetic complete galactose medium lacking uracil (panels D, E, and G). At time 0, the cultures were shifted to glucose medium for the indicated time. **(A–C)** Measurement of the sum of transcripts containing ITS1 and ITS2. Total RNA from BY4741, $P_{gal}$-uL4, and $P_{gal}$-uS4 was purified, and equal $A^{260}$ units were loaded onto a nylon membrane in a slot format (A). The membrane was probed with ITS1 oligonucleotide O1663 (B) or ITS2 oligonucleotide O1660 (C); see panel H for the position of the sequences to which they hybridize. **(D)** Analysis of 5′ ends at sites A2 and A3 in rRNA processing intermediates. $^{32}$P end-labeled primer O1345 (panel G) was hybridized to equal amounts of total RNA from $P_{gal}$-uL4 (YLL2083) and extended to 5′ ends of processing intermediates generated by cleavage A2 and A3 (see panel H). **(E)** Run-on analysis of rRNA transcription during repression of uL4 synthesis. Aliquots of cultures of $P_{gal}$-uL4 (YLL2083) growing in synthetic medium were harvested at the indicated times after the shift to glucose medium. Equal $A^{260}$ units from each sample were used for run-on labeling with $\alpha$-$^{32}$ATP. The products were hybridized to slot blots of pDK16 carrying the 35S rRNA transcription unit (Lindahl et al, 1994). **(F)** Yield of $A^{260}$ material per $OD_{600}$ units of culture harvested. Equal OD600 units of $P_{gal}$-uL22 and BY4741 were harvested at the indicated times. Total RNA was prepared by the hot phenol method and the $A^{260}$ of the final preparation was measured and normalized to time 0. **(G)** Pulse-chase analysis of rRNA processing during repression of uL4 synthesis. $P_{gal}$-uL4 (YLL2083) was grown in synthetic medium and harvested 0 or 6 h after the shift to glucose medium. The cells were labeled with $^{3}$H-uracil for 2 min and then incubated (chased) with a large excess of nonradioactive uracil for the indicated times. RNA was extracted and equal amounts of radioactivity from each sample were then fractionated by agarose gel electrophoresis, transferred to a membrane, and visualized by fluorography. **(H)** Map of the yeast Pol I rRNA transcription unit. The black boxes immediately below the map show the positions of the oligonucleotide probes used. The arrows labeled D, A2, A3, and C2 indicate cleavage sites relevant for this study. Other processing sites are omitted. The major rRNA processing intermediates (20S and 27S) are indicated below the map. For more details on rRNA processing, see Woolford & Baserga (2013).
Source data are available for this figure.

# Gene repressed for 16 hrs

## eS31 | eL43

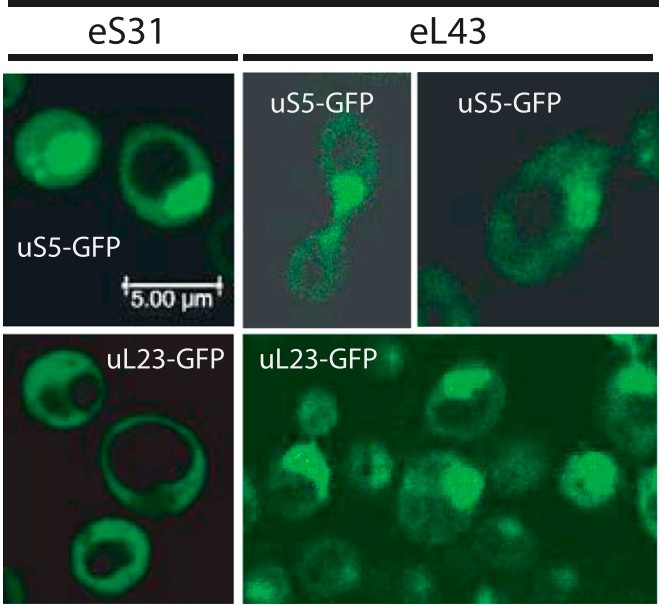

**Figure 7. Cellular localization of r-proteins during repression of a 40S or a 60S r-protein gene.**

P$_{gal}$-eS31 or P$_{gal}$-eL43, each carrying plasmid-borne genes for uS5-GFP or uL23-GFP expressed from their native promoters, were grown in synthetic galactose medium and shifted to synthetic glucose medium for 16 h. The cells were inspected by confocal microscopy. A uniform distribution of the GFP-tagged protein between the nucleus and cytoplasm indicates that the export of ribosomal precursor particles carrying the tagged protein is normal. A buildup of the GFP-tagged in the nucleus indicates that ribosomal precursor particles carrying the tagged protein are not matured to nuclear export competency.

(Fig 1B), the vacuoles do not appear to be a sign of immediately impending cell death. We speculate that vacuole formation may be the result of massive turnover of ribosomal components when complete assembly of ribosomal subunits is inhibited.

## Characterization of a 55S particle accumulating during cessation of 40S assembly

As mentioned above, a 55S peak develops after inhibiting 40S assembly. Sucrose gradients loaded with lysates prepared at different times after blocking uS4 or eS31 synthesis showed that the 55S peak appears 4–5 h after the shift to glucose medium and increases in size until 8 h after the shift (Figs 4D–I and S5A–F). The delay in 55S appearance suggests that it is not directly caused by the repression of 40S formation, but rather is a secondary effect. After 8 h, the 55S peak does not increase (compare Figs 4I with S2H and J), suggesting that the 55S particle is unstable such that its formation and degradation come to equilibrium after about 8 h. To determine if the accumulation of the 55S is affected by growth rates, we compared the height of the 55S peak after repressing the uS4 gene in cultures growing rapidly in rich YEP medium and relatively slower in synthetic medium (compare Figs S5G–J). The 55S peak is smaller compared with the 60S peak during the relatively slower growth in synthetic medium than in the YEP medium, indicating that growth rate affects the balance of synthesis and degradation of the 55S particle.

The Western analysis in Fig 4B and C shows that the 55S peak does not contain uS4 if the synthesis of either uS7 or uS4 is repressed. Thus, the absence of uS4 from the 55S peak cannot be due to its depletion as a result of blocking its synthesis. We, therefore, conclude that the 55S peak does not contain uS4 irrespective of which 40S r-protein gene is repressed. We further argue that because uS4 is important for the early steps in the formation of the small subunit (Kim et al, 2014), it is unlikely that the 55S particle is a defective 40S assembly particle. To confirm that it is a 60S-like particle, we analyzed the rRNAs in the 55S. RNA was purified from fractions of sucrose gradients loaded with lysates from cells harvested before and 8 h after repressing uS4 synthesis, and equal portions were fractionated by agarose gel electrophoresis. The RNA was transferred to a membrane and stained with methylene blue (Fig 8A and B). Comparison of the patterns from cells harvested 0 and 8 h after repressing uS4 synthesis shows that 8 h after the shift, the amount of 18S rRNA in the 40S region is diminished, as expected. More importantly, the RNA in the 55S–60S region is fragmented (indicated by red broken lines to the left of relevant lanes in Fig 8B). For further characterization, equal parts of the RNA purified from fractions in equivalent positions of the 0 h and 8 h gradients in an independent experiment were loaded in alternate slots of a northern experiment and probed with a mixture of oligonucleotides hybridizing to the distal part of ITS1 (O1680) and ITS2 (O1660), both parts of the 27S rRNA processing intermediates that are precursors for 25S rRNA (Fig 6H). The 27S pre-rRNA band was much stronger in RNA from the 8-h gradient compared with the 0-h gradient (Fig 8C). Furthermore, much more 27S rRNA sedimented with the 60S peak than the 55S peak. Because pre-ribosomal intermediates accumulate in proportion to their average lifetime (Lindahl, 1975), the relatively weak 27S signal in the 55S peaks could be interpreted to mean that 55S is a short-lived precursor in the 60S subunit assembly pathway, whereas the 27S signal in the 60S peak would represent longer living precursors, probably the canonical 66S pre-60S particles (Woolford & Baserga, 2013) that co-sediment with the mature 60S subunits under the conditions of our sucrose gradients. However, if this were the case, the height of the 55S peak in the A$^{260}$ trace should be much smaller than the height of the 60S peak, especially since most A$^{260}$ material in the 60S peak represents mature 60S particles. In contrast to this prediction, the height of the 55S peak was similar to that of the 60S peak (Figs 8A, B, 4B, and I), arguing against the idea that the 55S is a 60S precursor particle. Rather, we suggest that the 55S particle is a derivative of, and not a precursor for, the mature 60S subunit. We stripped the blot and probed it with an oligonucleotide hybridizing to the 5′ end of mature 25S rRNA (Fig 8D). Because the 60S peak is substantially larger in the 8-h gradient than in the 0-h gradient (even though the gradients were loaded with the same total amount of A$^{260}$ material), the intensity of the 25S bands in the 8-h samples to some degree occluded the bands in some of the 0-h fractions. We, therefore, excised selected lanes from the whole-blot image (indicated by dots at the bottom of the lanes) and adjusted the contrast to optimize viewing of individual lanes (Fig 8E). Comparison of both the whole-blot image and the optimized lane images together show that both the 60S and 55S peaks contain 25S rRNA fragments after, but not before, the cessation of 40S assembly (degraded 25A fragments are indicated by red broken lines in Fig 8A, B, and D). We suggest that the results in Fig 8 together indicate that free 60S

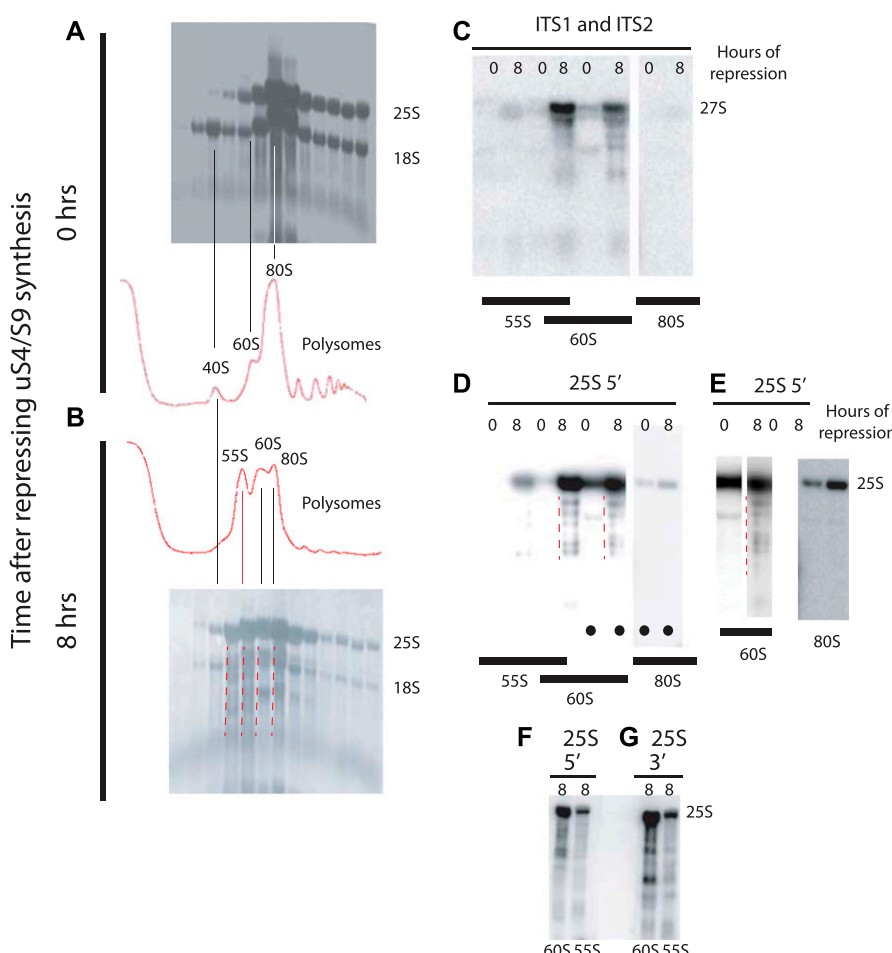

**Figure 8. Analysis of the 55S peak appearing after repressing the gene for 40S r-protein uS4.**

$P_{gal}$-uS4 was grown in YEP-galactose medium, and aliquots were harvested at 0 h and 8 h after the shift to glucose medium. **(A, B)** Equal $A^{260}$ units of whole cell extracts were fractionated on sucrose gradients and equal portions of the fractions sedimenting as 40S or larger particles were fractionated on an agarose gel, transferred to a membrane, and stained with methylene blue. **(C)** Equal portions of fractions from identical positions in a set of sucrose gradients from a biological replicate experiment were loaded into alternate slots of an agarose gel, transferred to a membrane, and probed with a mixture of equal radioactivity of end-labeled oligonucleotides O1680 and O1660 that hybridize to ITS1 (distal to the A2 processing site) and ITS2 (Fig 3H). **(D)** The membrane was then stripped of radioactivity and probed with an oligonucleotide specific to the 5′ end of 25S rRNA. The positions of the $A^{260}$ peaks in the gradients are indicated below the images of the membrane in (C, D). **(E)** Lanes marked with black dots in panel (D) were excised electronically from the image in (D) and optimized for viewing of the individual lanes. **(F–G)** Northern blots of RNA from the 55S and 60S fractions of a sucrose gradient of samples harvested 8 h after addition of glucose and probed with radioactive oligonucleotides hybridizing to the 5′ (panel F) or the 3′ (panel G) ends of 25S rRNA. Broken red lines to the left of the relevant lanes indicate rRNA degradation products.

subunits that fail to find a 40S partner become vulnerable to ribonuclease(s) and undergo a conformational change.

To determine if the fragmentation of the 25S rRNA was due to exo- or endonucleases, we compared the patterns of bands generated by probing for the 5′ and 3′ ends of the mature 25S (Fig 8F and G). If the fragments in the 55S and 60S peaks were due only to digestion by a single 3′ > 5′ exonuclease, such as the nuclear exosome, the 3′ probe should only hybridize to full length 25S rRNA. However, the 3′ probe hybridizes to both full-length and shorter fragments, and the pattern of the shorter fragments are different whether the blot was probed with the 5′ or the 3′ probe. This is not compatible with degradation by a single exonuclease attacking only from the 5′ or 3′ end. Rather, the degradation must be accomplished by an endonuclease or a combination of different nucleases.

## Discussion

### Disruption of assembly of one subunit affects the stability of the other

Ribosomes are the universal machines for protein synthesis and, thus, integral to the biological growth process. Inhibiting ribosome formation draws ripples in virtually all parts of the cell's metabolism (Shamsuzzaman M, Gregory B, Bruno V, and Lindahl L, in preparation). Among other effects, the cell cycle is arrested in early G1, and the cells undergo morphological changes (Bernstein et al, 2007; Thapa et al, 2013; Soifer & Barkai, 2014; Shamsuzzaman et al, 2017). In spite of these drastic effects, most of the cells remain viable for a long time during abrogation of ribosomal subunit formation (Fig 1B).

Here, we have investigated another consequence of disrupting ribosomal subunit assembly, namely, the distortion of a balanced number of the two ribosomal subunits. Ribosome biogenesis consumes a major fraction of the cell's resources, especially at rapid growth (Schaechter et al, 1958; Maaløe & Kjeldgaard, 1966; Warner, 1999). If this investment is to be used optimally, the two subunits must be made in equal numbers because they work in pairs. However, some mutations uncouple the formation of the two subunits: Depletion or inactivation of factors working before the cleavage of the primary rRNA transcript that separates the small from the large subunit-specific rRNAs thus affects the biogenesis of both subunits. However, after this cleavage event, ribosome biogenesis occurs through subunit-specific pathways. Although disruption of one of these pathways often delays the cleavage of the primary transcript, it does not significantly disrupt the throughput of the other (see the Introduction section and Fig 6G). Mutations in genes for r-proteins or factors specific to catalysis of only one of the

subunit pathways would, therefore, a priori lead to perpetual accumulation of an excess of one of the subunits. This imbalance could distort ribosome function. For example, an excess of one subunit could sequester translation factors, or an excess of 40S subunits could sequester mRNA in translation initiation complexes that cannot be converted to translating ribosomes because of the deficit in 60S subunits. It is, therefore, important to understand if there are mechanisms for rectifying an imbalance between the subunits.

We have addressed this question by examining the accumulation of the subunits and their components after conditionally abrogating the assembly pathway for one or the other of the subunits. Contrary to the expectation that unequal subunit accumulation is simply tolerated by the cell, we have found that inhibition of 60S subunit accumulation prevents accumulation of excess 40S subunits. We do not observe the reciprocal response; interference with assembly of the 40S subunit does not immediately prevent accumulation of the 60S subunit. However, the 25S rRNA of free 60S subunits becomes fragmented and a novel 55S particle derived from the 60S that appears after a few hours. This suggests that free 60S subunits may eventually turnover, albeit at a rate much lower than is the case for the excess 40S subunits.

While this manuscript was under review and revision, a publication appeared with proteomic data for strains lacking one of the paralogs for a given r-protein (Cheng et al, 2019). These authors concur with our conclusion that manipulation of 60S r-protein affects accumulation of r-proteins specific to both subunits, whereas reduced formation of 40S r-proteins affects accumulation of 40S r-proteins only.

### Post-assembly turnover of the 40S subunit

To understand how 40S accumulation is linked to 60S assembly, we asked if the 40S assembly is blocked or, alternatively, assembled subunits are degraded. Two lines of evidence show that the limitation of 40S accumulation occurs post assembly. First, quantification of the internal transcribed rRNA spacers and run-on transcription show that rRNA transcription relative to $A^{260}$ of cell extracts (total RNA) is not changed significantly (Fig 6B–E). Furthermore, the $A^{260}$ material per cell, as determined by the $OD_{600}$ of the culture, remains virtually constant (Fig 6F). Together, these observations suggest that the specific rate of 35S rRNA transcription (d(rRNA)/(total RNA)) does not change. Second, pulse-chase analysis showed that the 18S rRNA is efficiently processed to its final size (Fig 6G), whereas 25S and 5.8S rRNA are degraded. Because unassembled rRNA is typically unstable, this suggests that 40S subunits are assembled. However, we cannot exclude that modification of rRNA and r-proteins could be incomplete, even though 18S has reached its final size. We also note that initially, excess free 40S subunits do accumulate, suggesting that events secondary to the disruption of 60S assembly are necessary to destabilize the free 40S subunits. For example, the pool of free 40S subunits may have to increase before the chance of encountering a degradation enzyme becomes significant, or synthesis of degradation enzyme(s) may have to be induced.

The final cleavage in the maturation of 18S rRNA in the pre-40S particle occurs after export of the precursor particle in a complex

with a mature 60S subunit (Lebaron et al, 2012; Strunk et al, 2012; Garcia-Gomez et al, 2014). Thus, the turnover of excess free 40S subunits might be due to an insufficient number of mature 60S subunits to facilitate the final maturation of 40S. In addition, it is known that the concentration of ribosomal proteins is similar in the nucleus and cytoplasm during normal growth, but inhibition of the maturation of one of the subunits causes accumulation in the nucleus of proteins from this subunit (Hurt et al, 1999; Milkereit et al, 2003). Consistent with this, we observed nuclear accumulation of uS5 after repression of the eS31 gene (Fig 7). Unexpectedly, both the 60S protein uL23 and the 40S protein uS5 accumulated in the nucleus after repression of the uL40 (Fig 7) or uL4 genes (not shown), showing pre-40S export is linked, at least to some degree, to 60S assembly. Perhaps, pre-60S particles are necessary for complete modification of rRNA and/or proteins in the pre-40S subunit; under-modification could affect the stability and efficiency of nuclear export of late pre-40S particles. Potentially, the turnover of excess 40S could occur by a combination of inhibition of both nuclear export and final maturation in the cytoplasm. It is also known that mature 40S subunits containing mutated rRNA are degraded (Rodriguez-Galan et al, 2015), but there are no known mutations in the 18S sequence of our strains.

Together, these considerations lead us to speculate that pre-40S and perhaps even mature 40S subunits that fail to partner with a mature 60S subunit because of the 60S deficit may be vulnerable to cytoplasmic nucleases because the interface region, which is rich in rRNA loops, is exposed in the free subunits, but protected while it is paired with a 60S subunit (Ben-Shem et al, 2011).

### 60S subunits are destabilized when 40S assembly is blocked

Repression of 40S r-protein genes leads, with a delay of several hours, to fragmentation of a significant fraction of 25S rRNA in free 60S subunits but not in the 80S ribosome. Correlated with this is the appearance of a novel ribosomal particle (55S, Figs 4 and 8) derived from the 60S subunit. This particle has not previously been described, although it may be the reason for asymmetry of 60S peaks, trailing toward the top of the gradient, in several previous reports describing the effects of the depletion of 40S assembly factors (see e.g., Milkereit et al (2003); Perez-Fernandez et al (2007); Fernandez-Pevida et al (2016)). Furthermore, previous reports showed that mutations in the yeast genes TRM112 or BUD23, encoding factors necessary for efficient 40S assembly, generate a peak sedimenting similarly to the 55S seen in our experiments; this aspect of the mutant phenotypes was not discussed (Sardana & Johnson, 2012; Sardana et al, 2013).

The 55S peak did not increase after about 8 h, suggesting that equilibrium between formation and degradation of the particle is eventually established. Together with the fragmentation of the 25S rRNA, this indicates that the free 60S subunits become unstable after several hours of repression of 40S formation, but the turnover rate of 60S is clearly much lower than the degradation of 40S. This attack on the 60S most likely happens in the cytoplasm because 60S proteins do not accumulate in the nucleus after inhibition of 40S assembly (Fig 7), indicating that 60S assembly and nuclear export proceeds unimpeded.

Some of the fragmented rRNA in the 60S peak hybridizes to the transcribed spacer probes, suggesting that pre-60S particles with incompletely processed 25S rRNA are exported to the cytoplasm. Even though it is believed that 25S rRNA normally is fully processed before nuclear export of 60S subunits, precursor 60S particles containing ITS2 can escape to the cytoplasm during impaired ITS2 processing and that such cytoplasmic pre-60S particles form defective 80S ribosomes that are degraded (Sarkar et al, 2017). We speculate that the absence of 40S assembly can subtly disturb the nuclear maturation of 60S. In any case, the deficit of 40S subunits must lead to accumulation of free 60S subunits, both mature and, perhaps especially, immature particles. As suggested above for the 40S subunit, the interface side of unpaired 60S could be vulnerable to nucleases. This interpretation is supported by the fact that the 25S rRNA in the 80S ribosomes appears to be intact (Fig 8E). Another possibility is that the 60S may be under-modified in the absence of 40S assembly and, therefore, subject to degradation, for example, by the pathways that eliminate 60S particles with mutations in the peptidyl-transferase center (Cole et al, 2009).

### Variation in the rates of degradation of uL18 after repression of different 60S r-protein genes

The uL18 concentration decreases more slowly during repression of uL4 synthesis than during inhibited eL40 gene expression (Fig 5B and D). This may be related to uL18 forming a complex with uL5, 5S rRNA, and two 60S assembly factors before its incorporation into pre-60S particles late in the assembly process (Zhang et al, 2007). Repressing the uL4 gene aborts the 60S assembly at an early step of the assembly pathway (Gamalinda et al, 2014), which may prevent transfer of uL18 to a pre-60S particle. Therefore, uL18 may remain in the pre-incorporation complex where it may be semi-protected from degradation. In contrast, eL40, like uL18, is a late assembly protein, so interrupting the eL40 synthesis may stop the assembly after uL18 has been transferred to the pre-60S particle and, thus, make uL18 as vulnerable as the rest of the pre-60S particle.

### Implications

We hypothesize that the linking of 40S accumulation to 60S subunit formation has evolved to secure proper function of the translation process. A surplus of 40S subunits would lead to formation of translation initiation complexes that cannot be converted to translating 80S ribosomes because of the shortage of 60S subunits. Potentially, this could indiscriminately sequester mRNAs and distort the synthesis of many proteins. Excess free 60S subunits would not have a similar effect on translation.

Our findings may also be relevant to diseases caused by mutations in genes for r-proteins and factors for ribosome assembly and rRNA modification (Mattijssen et al, 2010; Narla & Ebert, 2010; Tafforeau et al, 2013; Danilova & Gazda, 2015; Farley & Baserga, 2016; Bustelo & Dosil, 2018). Because ribosomal assembly in yeast shares many aspects of human ribosome formation (Tafforeau et al, 2013), our work suggests that the mechanism for such diseases may involve interactions between subunit assembly and stability. Experimental protocols involving many hours of inhibition of ribosomal assembly are subject to the criticism that the extended time

allows development of secondary reactions not directly related to the initial assault (e.g., Kos-Braun & Kos (2017)). However, in the context of genetic diseases, the mutational stress is permanent, making it relevant to investigate the long-term effect of disturbing ribosome assembly.

During normal growth, the fraction of the cell mass constituted by ribosomes increases with growth rate, which has led to the dogma that ribosome formation is optimized for minimal drain on cell resources (Maaløe & Kjeldgaard, 1966; Warner, 1999). Indeed, signaling pathways emanating from target of rapamycin (TOR) repress transcription of ribosomal genes during poor nutritional conditions or oxidative stress (Claypool et al, 2004; Mayer & Grummt, 2006; Philippi et al, 2010). However, our results show that preservation of resources is not always the cell's priority. Transcription of rRNA and r-protein genes continues when assembly of one of ribosomal subunits is abrogated and excess 40S subunits, and likely excess 60S subunits, are degraded. When RNase MRP, a factor required for formation of both subunits, is inactivated, rRNA transcription also persists, but processing of both 18S and 25S is unsuccessful (Lindahl et al, 2009). Moreover, r-proteins that fail to be incorporated during distortion of the assembly of one of the ribosomal subunits are turned over (Gorenstein & Warner, 1977). In all cases, there is a massive turnover of ribosomal material rather than a reduction in synthesis and assembly of ribosomal components, implying a large waste of cell resources.

During lack of nutrients or oxidative stress, TOR is cued to reduce transcription of ribosomal genes and induce accumulation of a 23S rRNA transcript (Kos-Braun & Kos, 2017). However, we did not see such accumulation of 23S rRNA during repression of uL4 synthesis (Fig 6G), suggesting that TOR regulation may not be involved during unbalanced assembly of subunits. Perhaps, TOR is deactivated by stress originating from sources external to the cell, but not by stress emanating from internal sources such as mutations in genes for ribosomal components or assembly factors.

## Materials and Methods

### Nomenclature

We used the universal nomenclature for r-proteins, but classic yeast nomenclature is also shown in figures (Ban et al, 2014).

### Strains and growth conditions

Strains are listed in Table S1. In each strain, the indicated r-protein or ribosome assembly factor was expressed only from a gene transcribed from the *Gal1/10* promoter. We refer to these strains as $P_{gal}$-xx, where xx is the name of the protein expressed from the *GAL* promoter. Many r-proteins in yeast are encoded by two paralogous genes. With the exception of r-protein uL4, we only analyzed repression of one of the two paralogues, but because abrogating the synthesis of different proteins from a given subunit generated the same pattern of ribosomal subunit accumulation, it is unlikely that the particular r-protein allele used has any important effect on the results. For repression of uL4 synthesis, all experiments, except in

Fig 6D–E and G, were carried out with JWY8402 expressing the gene encoding uL4A (RPL4A) from the galactose promoter. In Fig 6D–E and G, the gene encoding the uL4B allele (*RPL4B*) was under galactose control (strain YLL2083).

Plasmids carrying C-terminal fusions of genes for GFP and uS5 or uL23 expressed from the native r-protein promoters were described previously (Hurt et al, 1999; Milkereit et al, 2003). To enable induction of uL23-GFP synthesis during repression of uS17/S11 synthesis, a DNA fragment harboring the gene for the artificial *β*-estradiol–sensitive transcription factor Z3EV (McIsaac et al, 2013), natMX6, and a fragment of *ADE2* was integrated into the chromosomal *ADE2* gene of Pgal-uS17 (Y325) by selecting for nourseothricin resistance (Hentges et al, 2005). Next, a gene for GFP-tagged uL23/L25 was placed under control of the Z3EV-responsive promoter PNM3 and integrated into the URA3 gene using 5-fluororotic acid for selection.

Steady-state cultures growing with shaking at 30° in YEP-galactose or galactose synthetic complete medium (Sgal) lacking uracil were shifted to pre-warmed YPD medium or glucose synthetic complete medium (SD) lacking uracil (Sherman et al, 1979). Aliquots for analysis of ribosomes or ribosomal components were withdrawn immediately before (time 0) and at the indicated times after the shift. The cultures were diluted with pre-warmed medium whenever necessary to keep the $OD_{600}$ below 0.8. Cell density was measured in a 10-mm cuvette using a Hitachi U1100 spectrophotometer (Hitachi High-Technologies Corporation). $OD_{600}$ = 1 corresponds to $1.5–2 \times 10^7$ cells (colony-forming units) per milliliter.

## Western analysis

Cells were spun down at 5K for 10 min, washed with and suspended in buffer A (20 mM Tris–HCl, 150 mM KCl, 5 mM Na-EDTA, 0.1% Triton X-100, and 10% glycerol), dithiothreitol (1 mM), and phenylmethylsulfonyl fluoride (20 μl/ml). The cells were then lysed by vortexing for 15 min at 4°C with glass beads. Equal numbers of $A^{280}$ units from three independent cultures for each time point were applied to 15% polyacrylamide gels. Samples from all time points were fractionated on the same gel. After electrophoresis, proteins were transferred onto PVDF membranes for 1.5 h at 130 V using the Trans-Blot cell apparatus (Bio-Rad). The membranes were then incubated in a blocking solution (20 mM Tris, pH 7.5, and 150 mM NaCl, 0.1% Tween 20, 5% dry milk [LabScientific, Inc.]) before incubating with primary antisera (1:10,000–1:4,000 dilutions). Rabbit polyclonal antisera for the yeast ribosomal proteins were prepared for our laboratory by Covance using synthetic peptides with the sequence of 20–22 N-terminal amino acids of uS4, uL4, uL5, and uL18 as antigens. The membrane was cleared of excess antisera with washing buffer (25 mM Tris-Cl [pH 7.4], 2.5 mM KCl, and 200 mM NaCl) before it was incubated with goat anti-rabbit IgG (H+L)-AP conjugate secondary antibody (Bio-Rad). Bands were detected by exposing the membrane to Amersham ECF substrate (GE Healthcare) followed by scanning on a Storm 860 Imager System (Molecular Dynamics). Bands were quantified with ImageJ, averaged for a given time point, and normalized to the average of three 0-h samples. The normalized values were plotted, together with the respective standard errors of the mean. Finally, the values for uS4, uL5, and uL18 were normalized to the value of uL4 in the same lane, averaged for each time point and plotted together with standard errors of the mean. Antisera were titrated against $A^{280}$ units loaded and used in excess of the r-proteins in the samples for determination of r-proteins in the experimental samples (Fig S6).

## Sucrose gradient analysis

In many reports using sucrose gradients for ribosome fractionation, cycloheximide was added to the growing culture before harvest. However, short incubations with the drug artificially increase the fraction of ribosomes in polysomes and the level of "halfmers" (polysomes with a number of 80S ribosomes plus an initiating 40S waiting for a 60S to join the complex) (Helser et al, 1981). This presumably reflects the drug's blocking of ribosome run-off from mRNAs, whereas not affecting translation initiation. The change in the balance of initiation and run-off also affects the accumulation of halfmers, which explains why we see fewer halfmers than those reported by others after inhibition of 60S assembly. Because the cycloheximide effects on the sucrose gradient pattern could be considered an artifact, we did not add cycloheximide to the culture but did include it in the lysis buffer to preserve polysomes after lysis. The cells were quick-chilled over ice, spun down, and washed with ice-cold water. The pellet was then resuspended in ice-cold gradient buffer (50 mM Tris acetate, pH 7, 50 mM $NH_4Cl$, 12 mM $MgCl_2$, and 1 mM DTT) containing 50 μg cycloheximide per ml. The mixture was transferred to a tube (NC9437081; Sarstedt) containing 2.5 g glass beads (G9268 or G8772; Sigma-Aldrich) and vortexed five times at a maximum speed for 30-s intervals at 4°C. The resulting cell lysate was spun down twice at 13,000 g for 15 min at 4°C, and the pellet was discarded after each spin. Equal $A^{260}$ units of the supernatant after the second spin were loaded onto 10–50% sucrose gradients in gradient buffer. The gradients were spun at 40,000 rpm for 4 h at 4°C using an SW40Ti Beckman rotor. Fractions (500 μl) were collected using an ISCO Foxy Jr fraction collector, pumping at 1 ml/min. Gradients that were compared were loaded with the same number of $A^{260}$ units of lysate, and the sensitivity of the flow-colorimeter was set the same for gradients to be compared. The relative amounts of $A^{260}$ material in the different peaks were determined using ImageJ.

## RNA analysis

### Total RNA extraction
Cells were harvested by centrifugation and stored at –80°C. The cells were then resuspended in ice-cold water, and RNA was extracted using the phenol–chloroform method as described (Lindahl et al, 1992). Alternatively, RNA was prepared using a RiboPure kit (Life-Technologies AM1926) following the manufacturer's instructions.

*Extraction of RNA from RNA polysome gradient fractions* was performed as described (https://case.edu/med/coller/Coller%20Protocol%20Book.pdf) (Cigan et al, 1991; Nielsen et al, 2004). Briefly, individual sucrose gradient fractions were mixed with an equal volume of ethanol and precipitated overnight at –80°C. The pellets were resuspended in LET (25 mM Tris–HCl, pH 8.0, 100 mM LiCl, 20 mM Na-EDTA) and 1% sodium dodecylsulfate, extracted twice with one volume phenol/chloroform/isoamyl alcohol (25:24:

1), and precipitated overnight at 80°C using 2.4 volumes ice-cold ethanol and 1/5 volume 10 M ammonium acetate.

### Ribosomal RNA analysis

(i) The ratio between 18S and 25S rRNA was measured by fractionating 3 $\mu$g (0.06 A$^{260}$ units) of total RNA on 0.8 or 1.2% agarose gels that were stained with ethidium bromide after the run. (ii) ITS1 and ITS2 were quantified by "slot blot analysis" as follows. 3 $\mu$g total RNA in sterile water were deposited directly onto an Amersham Hybond-N nylon membrane (GE Healthcare Life Sciences) using the Minifold II Slot-Blot system (Schleicher & Schuell). After cross-linking, the membranes were hybridized with $^{32}$P-end–labeled probes (Sambrook et al, 1989) complementary to segments of ITS1 or ITS2 (Fig 6H) in a solution made from 20 ml 100X Denhardt's solution (http://cshprotocols.cshlp.org/content/2008/12/pdb.rec11538.full?text_only=true), 60 ml 20× SSC, 120 ml H$_2$O, and 2 ml 10% SDS. The membrane was prehybridized with rotation in the solution without the radioactive probes for at least 1 h. Radioactive probes were then added (5 × 10$^6$ cpm per chamber) after which the mix was incubated with rotation at 37° overnight. Finally, the blot was washed three times for 10 min at room temperature with 6xSSC and 0.01% SDS and exposed to a storage phosphor screen (Molecular Dynamics) for 4 h. Bands were detected by scanning on a Storm 860 or Typhoon 9200 Imager (Molecular Dynamics). We note that the RNA preparations were not treated with DNase and could, therefore, be contaminated with rDNA. However, this would not significantly affect the results because there are only ~150 copies of the rDNA per genome (Dammann et al, 1993), whereas approximately 1,000 rRNA transcripts are made per minute (the cell contains about 200,000 ribosomes [von der Haar, 2008] and the doubling time is ~150 min). Because the ITS1 and ITS2 parts of the transcript have a lifetime of 5–10 min (Fig 6G), the cell contains about 5,000–10,000 copies of the internal transcribed spacers in primary and processing intermediates, far exceeding the number of rDNA copies. (iii) RNA purified from sucrose gradient fractions was fractionated by gel electrophoresis on 0.8% agarose gels in 0.5xTBE (45 mM Tris-borate, pH 8.3, and 1 mM EDTA-Na) and was transferred onto an Amersham Hybond-N nylon membrane (GE Healthcare Life Sciences) using a Model 230600 Boekel Vacuum Blotter for 2 h at ~40 mbar. The RNA was then cross-linked to the membrane and stained with 0.04% methylene blue in methanol. Next, the membrane was destained with 25% ethanol and hybridized with the indicated $^{32}$P-end–labeled probes as above.

Sequences of the oligonucleotide probes used are:

O1345: TGGAATACCAAGGGGCGCAATGTG
O1663: CTCTTGTCTTCTTGCCCAGTAAAAG
O1660: AGGCCAGCAATTTCAAGTTAACTCC
O1680: CCAGTTACGAAAATTCTTGTTTTTGAC
5′ 25S: ACTCCTACCTGATTTGAGGTCAAACC
3′ 25S: GGCTTAATCTCAGCAGATCGTAAC

See also Fig 6H.

### Pulse-chase labeling

Pulse-chase labeling was performed essentially as described by Dunbar et al (2000). Briefly, YLL2083 was transformed with an "empty" Ycplac33 (URA3) plasmid and grown to an OD$^{600}$ of ~0.6 in SGal–Ura. Half the culture was then diluted 1:20 into fresh SD–uracil and both parts were incubated for an additional 6 h while keeping the OD$^{600}$ under ~0.8. Approximately 50 OD$^{600}$ units of the cells in SD–uracil and 10 OD$_{600}$ units of cells in SGal medium lacking uracil were resuspended in 0.4 ml SD or SGal lacing uracil, respectively, and labeled with 25 $\mu$Ci of [5,6-$^3$H] uracil per milliliter (47 Ci/mmol) at 30°C for 2 min. The use of a larger number of cells in the glucose culture was necessary to obtain approximately equivalent incorporation of $^3$H in the galactose and glucose cultures. This was expected because the substantial turnover of 25S rRNA reduces the net drain on the pyrimidine triphosphate pools and, therefore, slows the increase of the specific activity of the triphosphate pools. The chase was initiated by adding 4 ml of medium containing 1 mg/ml of nonradioactive uracil to the labeled cells. At 0, 5, 10, and 20 min after the chase, 1-ml aliquots of cells were pelleted, resuspended in ice-cold dH$_2$O, re-pelleted, and flash-frozen in a dry ice-ethanol bath. Total labeled RNAs were isolated by the hot phenol glass bead technique (Lindahl et al, 1992). Equal amounts of radioactivity from each of the purified RNA samples (30,000 cpm) were electrophoresed on either a 1.2% agarose gel for monitoring larger fragments or an 8% polyacrylamide gel for low molecular weight rRNAs. Labeled RNAs were transferred to a Zeta-Probe membrane (GE Healthcare Life Sciences), sprayed with EN3-HANCE (Dupont), and exposed to X-ray film.

## Cell viability

The total number of cells was counted under the microscope using a hemocytometer. Connected cells that had not completed cell separation were counted as one potential colony-forming unit. Viable cells (colony-forming units) were quantified by plating on YEP-galactose medium, incubating at 30° for 2 d, and counting the colonies. The experiment was repeated three times for each strain in both galactose and glucose media. The average from all three experiments and the standard error of the mean are shown.

### Other methods

The bands on northern and Western images were quantified and analyzed using ImageJ software. Run-on transcription was performed as described (Elion & Warner, 1986).

### Reproducibility

Experiments were performed two or more times. Error bars indicating the standard error of the mean based on samples from three independent cultures (biological repeats) are shown in Figs 3, 5, 6B, and C. To determine if rRNA transcription and 40S assembly continues after repression of uL4 synthesis, three different types of experiments generating mutually supportive results were each performed once (Fig 6D, E, and G). Note that these experiments are consistent with the results of yet another approach in Fig 6B and C, which includes statistical treatment of triplicate experiments. Moreover, Fig 6G generated the same result as pulse-chase labeling after depletion of r-protein uL1/L1 (data not shown), another r-protein specifically required for assembly of the 60S, but not the 40S subunit (Deshmukh et al, 1993).

# Supplementary Information

# Acknowledgements

This work was supported by grant 0920578 from the National Science Foundation and a grant from the Provost's Office, University of Maryland, Baltimore County (UMBC). B Gregory was supported by NIH IMSD training grant 5R25GM055036 and the Meyerhoff Graduate Program at UMBC. We thank John Woolford (Carnegie Mellon University), Philipp Milkereit (University of Regensburg), Ed Hurt (The University of Heidelberg), Mercedes Dosil (CSIC-University of Salamanca), R Scott McIsaac (Princeton University), and David Botstein (Princeton University) for strains. We also thank Henrik Nielsen and Nicolai Krogh (University of Copenhagen) for their comments on the manuscript and stimulating discussions, and Jelena Jakovljevic for discussion of polysome preparations. We also thank Benedikte Traasdahl for help with the manuscript.

## Author Contributions

B Gregory: conceptualization, data curation, formal analysis, investigation, methodology, and writing—original draft, review, and editing.
N Rahman: conceptualization, data curation, formal analysis, investigation, methodology, and writing—original draft, review, and editing.
A Bommakanti: conceptualization, data curation, formal analysis, and writing—original draft.
M Shamsuzzaman: conceptualization, formal analysis, investigation, and methodology.
M Thapa: data curation and formal analysis.
A Lescure: data curation.
JM Zengel: conceptualization, funding acquisition, investigation, project administration, and writing—original draft, review, and editing.
L Lindahl: conceptualization, formal analysis, funding acquisition, investigation, visualization, methodology, project administration, and writing—original draft, review, and editing.

## Conflict of Interest Statement

The authors declare that they have no conflict of interest.

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
