## [Reviewer comments · Life Science Alliance]

Life Science Alliance

The small and large ribosomal subunits depend on each other for stability and accumulation

Brian Gregory, Nusrat Rahman, Ananth Bommakanti, Md Shamsuzzaman, Mamata Thapa, Alana Lescure, Janice Zengel, and Lasse Lindahl

DOI: <https://doi.org/10.26508/lsa.201800150>

Corresponding author(s): Lasse Lindahl, University of Maryland, Baltimore County

Review Timeline:	Submission Date:	2018-08-08
	Editorial Decision:	2018-09-06
	Revision Received:	2019-01-17
	Editorial Decision:	2019-02-05
	Revision Received:	2019-02-21
	Accepted:	2019-02-25

Scientific Editor: Andrea Leibfried

Transaction Report:

September 6, 2018

Re: Life Science Alliance manuscript #LSA-2018-00150-T

Lasse Lindahl
University of Maryland, Baltimore County
Biological Sciences
Baltimore 21250

Dear Dr. Lindahl,

Thank you for submitting your manuscript entitled "The small and large ribosomal subunits depend on each other for stability and accumulation" to Life Science Alliance. The manuscript was assessed by expert reviewers, whose comments are appended to this letter.

As you will see from the comments, all three refs state appreciate that your study offers deeper insight on how subunit stoichiometry is balanced during ribosome biogenesis. However, they also raise a number of technical and conceptual concerns that will have to be addressed before they can support publication of your manuscript here.

Ref #1 points to technical problems with the system used - this needs to be addressed/clarified. The other concerns from this referee can largely be addressed with text changes
Ref #2 requests additional controls and quantification and points to potential issues with experimental conclusiveness that should be addressed. I would also ask you to clarify the concerns raised about rRNA quantification using slot blots.
Ref #3 raises substantial - but in our view very constructive - concerns about the overall experimental setup and the conclusions that can be derived from it. In addition, this referee points to a number of missing controls or potentially inconclusive experiments that could undermine the main message of the study. We realise that ref #3 asks for a lot of additional work but in our view these are valid and constructive points that should all be addressed.

Should you be able to address these criticisms in full, we could consider a revised manuscript. I should remind you that it is Life Science Alliance policy to allow a single round of revision only and that, therefore, acceptance or rejection of the manuscript will depend on the completeness of your responses in this revised version. I do realize that addressing all the referees' criticisms will require a lot of additional time and effort and be technically challenging. I would therefore understand if you wish to publish the manuscript rapidly and without any significant changes elsewhere, in which case please let us know so we can withdraw it from our system.

Thank you for this interesting contribution to Life Science Alliance. We look forward to receiving your revised manuscript.

Sincerely,

Andrea Leibfried
Executive Editor
Life Science Alliance

- A letter addressing the reviewers' comments point by point.
- An editable version of the final text (.DOC or .DOCX) is needed for copyediting (no PDFs).
- High-resolution figure, supplementary figure and video files uploaded as individual files: See our detailed guidelines for preparing your production-ready images, <http://life-science-alliance.org/authorguide>
- Summary blurb (enter in submission system): A short text summarizing in a single sentence the study (max. 200 characters including spaces). This text is used in conjunction with the titles of papers, hence should be informative and complementary to the title and running title. It should describe the context and significance of the findings for a general readership; it should be written in the present tense and refer to the work in the third person. Author names should not be mentioned.

B. MANUSCRIPT ORGANIZATION AND FORMATTING:

Full guidelines are available on our Instructions for Authors page, <http://life-science-alliance.org/authorguide>

Reviewer #1 (Comments to the Authors (Required)):

In this manuscript the authors have used a number of biochemical approaches to address the question whether 40S and 60S assembly/accumulation of ribosomal subunits is interdependent. The results show that blocking 60S subunit synthesis impairs 40S accumulation and nuclear export, however, blocking 40S synthesis does not significantly impair 60S biosynthesis or export. The data indicate that this is independent of rRNA transcription and that the observed decrease in 40S subunits is most likely the result of degradation of excess 40S. The idea that 40S and 60S assembly/accumulation is coupled (to some extent) is not novel, however, this relationship has not been so carefully examined up to now. Overall the data support their conclusions but I have a few suggestions that may improve the manuscript.

Comments:

Page 4: The authors discuss on page 4 that depletion of LSU proteins results in a decrease of 60S subunits but 40S levels were increased relative to other peaks. They conclude from this that 40S subunits were still produced. From these data alone I'm not sure I would draw this conclusion as there is no data showing the effect of 60S depletion on pre-40S rRNA processing. But later on page 9 they show nice data that suggest that 40S accumulation is in fact inhibited in response to blocking 60S biogenesis. For me this was a bit confusing. I am wondering if the authors could move Figure 4 into Figure 2 and I would also recommend adding Supplementary Figure 2 to the main figures as this is really an excellent control experiment. I would even consider adding an extra figure showing (lack of) incorporation of a GFP-tagged SSU r-protein. I think moving the figures around might help to improve the flow of the manuscript as it is then immediately clear to the reader what the effect of the depletions have on 40S and 60S assembly. Then the authors could then discuss the 55S particles.

Page 9: The following sentence is quite lengthy and a bit difficult to follow. I would recommend rephrasing this:

"In contrast to the pan-subunit effect of repressing 60S protein synthesis, restraining the synthesis of the 40S proteins uS4, uS10, or uS11 generated a subunit-specific response, i.e. the specific concentration of uS4 protein declined, while the abundance of 60S proteins uL4, uL5, and uL18 changed little (Figure 4E-G)."

Figure 5. The first two panels show graphs of ITS1 and ITS2 abundance, yet the header above the graphs says "ITS1 abundance". Should this not be "ITS abundance"?

If I would be really picky, I would argue that ITS1 abundance slightly decreases due to r-protein depletion. It is not statistically significant for most time-points but there is an effect. Perhaps the authors could state here that there is a mild effect on ITS1 accumulation but there clearly isn't any indication that rRNA transcription is significantly affected. The run-on experiment is important but the figure shows only a few slices of blots and there is no quantification/normalization. It would be very helpful if the authors could quantify the run-on results and provide error-bars.

Figure 6. Here the authors have used GFP-fusion r-proteins to look at export of ribosomal subunits. Why did the authors look at cells after 16 hours of gene repression? This seems quite long and, judging from the vacuoles, the cells are on their way out. Also, to be able to have a complete picture of the data, would it not be helpful to also show the localization of not repressed cells?

Finally, just a warning, we have had serious problems in the lab with this r-protein GFP reporter system. We noticed that when we integrated the GFP into the genome downstream of r-protein genes we often got a completely different result. Secondly, we have had cases where the rpS-GFP reporter showed clear nuclear accumulation, whereas FISH data showed clear cytoplasmic accumulation of 20S. Unfortunately, negative data is very difficult to publish...

I obviously won't insist that the authors do these experiments for this manuscript because they may have had different (more positive) experiences, but in the future I would strongly recommend using FISH to get a read-out of nuclear export of pre-ribosomes.

Discussion page 13. "Since assembly of the two ribosomal subunits occurs along separate largely independent pathways (Woolford and Baserga 2013), we were surprised that inhibition of the 60S subunit prevents accumulation of the 40S subunit."

I do not find this very surprising as it has been shown in several cases that depletion of LSU assembly factors can lead to 18S pre-rRNA processing defects. Perhaps a classical example is Rrp5 and I believe some of the HEAT repeat proteins show the same phenotype. Also the co-transcriptional processing model is not compatible with the idea that the processes happen independently. For example, co-transcriptional processing at site A2 requires Pol I to transcribe at least parts of the 25S sequence. It has been estimated that ~70% of the pre-rRNA is co-transcriptionally processed. Perhaps this would be worth highlighting this in the discussion as well. It might also be worth discussing the half-lives of the individual subunits. Is the 40S generally more stable than 60S? Could this partially explain why after 8h of depletion so much 40S is still present?

Reviewer #2 (Comments to the Authors (Required)):

The presented manuscript addresses the question of if and how cells ensure that equal levels of the small and large ribosomal subunits are produced in situation when ribosome biogenesis is disrupted. The authors describe an interesting observation that inhibition of either small (40S) or large (60S) ribosomal subunits synthesis pathways does not lead to the same outcome. While inhibition of the 60S synthesis reduced also accumulation of the mature 40S subunits in their experiments, the opposite was not true. Interestingly, authors' experiments suggest that 40S synthesis continues and reach perhaps partial maturity (at least based on presence of 18S rRNA), however, these subunits are perhaps not fully exported to cytoplasm. The data presented provide some insight into these intriguing questions. However, there is a number of issues in the current version that need to be answered.

1) Figure 1. A block in the 60S synthesis typically leads to the appearance of halfmer peaks in the polysomes (polysomes, where the 40S initiation complex is bound and waiting for the 60S to join). Could author provide on their explanation why it is not the case here (at least not at the 2.5h time-point in the Fig 1D)? Does the lack of halfmers indicate that the translation initiation was arrested?

2) Also, it would be informative to show the growths curves during the depletion for all the relevant figures. Were the cells still growing at 8h or 16h time points? Does the reduction of the total ribosome content correspond to the "dilution" of ribosomes by cell division or does it indicate an

active degradation/turnover process of the remaining ribosomal subunits (perhaps linked to the increased vacuoles)?

3) Figure 1. It is not clear if the presented polysome profile curves were normalized to the total A260 signal (clipping of the signal in the figure 1G and 1H indicates this was not done).

4) There is no visible green circle for the uL4 levels at 16h in the Figure 1K and 1L. Is it overlapping (then change the figure to show it) or missing?

5) Figure 3. The panel A, methylene blue staining - the quality of the figure is not great. The 18S seems to present in all the fractions at time 0, which should not be the case. There should be at least 1 fraction gap (e.g. in this case no signal in fractions 13 and 14). This indicates trailing of the 40S/18S in the gradient (or mixing during the collection of fractions) and prevents any meaningful interpretation. In addition, the presence of strong degradation smear in the critical lanes 8 and 10 could easily obscure any weaker signal, again preventing solid conclusion whether any 18S is or not present in the 55S peak. Ideally the quality should be improved, or the membrane probed also with a radioactive 18S probe.

6) Figure 3B,C. Why is there no signal for 27S and 25S rRNAs for the time 0 lanes, especially in the lane 11 corresponding to 60S peak? Both 25S and 27S must be present at time 0. This pattern does not correspond at all to the methylene blue staining, where 25S rRNA is clearly present in all "0h" lanes.

7) Figure 3D and 3E. and the last sentence of the second paragraph, page 8: "Thus, the fragmentation must involve endonucleolytic attacks". This is an important experiment, but the conclusion is not fully correct. The 5'- and 3'-end probes label a different pattern of bands. Such pattern can arise also from a mixture of 5'>3' and 3'>5' exonucleases and strictly speaking does not have to be made by an endonuclease. Please correct the statement

8) Figure 4. The authors show behavior of few other L proteins during L protein depletion, indicating coregulation/co-degradation of L proteins (as expected). Could they also show the behavior of other S proteins during an S protein depletion? Are also S proteins coregulated. Also, the charts miss the X-axis labelling.

9) Figure 5A-C,E. Pre-rRNA abundance. Authors claim that pre-rRNA abundance is increased during depletion. However, the experimental setup does not necessarily support this conclusion. Authors used a slot blot, where same amount of total RNA was deposited in each slot. There are two major issues with this:

a. The total RNA in later time points contains significantly less of mature ribosomal RNA (which in growing yeast cells represents 80% of total RNA). In the later time points of the depletion, when the total ribosome levels are strongly reduced, the total RNA will in proportion contain much more of non-mature rRNAs, including more of the pre-rRNAs. It thus leads to overloading of the remaining RNAs and artificially increasing the amount of precursors, obscuring the well-known and established reduction of the rRNA transcription per cell. These experiments need to be done by loading the same number of cells. Otherwise, only a qualitative statement that some (most likely strongly reduced) transcription continues can be made. The same applies to the run-on experiments and pulse-chase when same amount of total RNA is loaded.

b. Loading total RNA in a slot blot also prevents distinguishing between different pre-cursors. The ITS1 and ITS2 probes are complementary to several different precursors, such as the primary transcript 35S pre-rRNA (which will be detected by both probes!). It is well established that the 35S

levels increase during aberrant ribosome biogenesis, but not necessarily due to more transcription but due to stabilization/arrest of processing. In addition, both probes will also detect potential contaminating genomic DNA (which contains 200 repeats of rDNA) contamination. Therefore results from the slot blot these results are inconclusive.

10) Figure 3E. In the discussion section authors refer to this figure to show presence of 23S pre-rRNA. However, there is no label for the 23S band. While I can see some smear/weak bands in that are, it is not clear to which authors refer. Please include an arrow for the 23S band if clearly present and if not, correct the statement in the discussion.

Reviewer #3 (Comments to the Authors (Required)):

In this manuscript Gregory et al use a combination of Northern and Western analyses and immunofluorescence to study what happens to both ribosomal subunits when assembly of one or the other is blocked for extended times. This manuscript recapitulates observations known for a long time in the literature and combines them with some more in depth analysis. Overall this manuscript requires a more consistent analysis, overstates much of the data and is also incorrect in several instances. Details are below.

Major points:

1. Given that none of the observations in this manuscript are new, the strength would lie in a careful analysis. However, this is not done at nearly any point. For starters, the authors set themselves up poorly by switching everything from galactose to glucose. This switch from a so-so nutrient source to their favorite food affects ribosome assembly, and the analysis herein, but is not considered anywhere. A better way to do this would have been to use the strains they use, and then either have or not have plasmids encoding the ribosomal proteins under constitutive promoters, but analyze everything in glucose. While it might not be necessary to repeat all of the analyses in here, a subset of the experiments needs to be repeated using this cleaner system to deconvolute effects. This is minimally true for the transcriptional analysis, and the analysis of rRNA levels.
2. Figure 1, Figure 2, Figure S1 and Figure S3 show sucrose gradients for depletion of various small and large subunit proteins. But there is no consistency. For two proteins an entire time course is shown, but the two time courses are different. Then, for many the t=0 time point is missing. This is critical, because Gal overexpression of some proteins is actually detrimental. The authors should provide time courses, including minimally a 0, 6 or 8h and 16h timepoint for all RPs.
3. In Figure 2A&B, they want to draw conclusions about the 55S peak not having uS4, when uS4 is depleted. This is clearly not a valid conclusion. In addition, there is concern that in panel C, uS4 is not found in the 40S peak.
4. Figure 3 is very confusing. Why is there essentially no signal for the 25S 5' in the 0h depletion? This probe should pick up mature 25S, of which there is at least as much in the undepleted cells. Also, panels D and E require a control for undepleted sample. Furthermore, this Figure shows that their gradients are not able to effectively separate anything from anything as both 18S and 25S are all over the gradient. This must be repeated in a cleaner way, as it also casts serious doubt on what the 55S complex actually is. Finally, the authors claim that Figure 3 shows that there is no 18S rRNA in the 55S peak. It is totally unclear how one could make such a claim. These lanes are so smeary from the degradation products observed that one would never be able to see 18S rRNA. If the authors want to make such a claim, they must do it by Northern probing, with at least 2 probes at different locations to rule out degradation of 18S.
5. Figure 4 is potentially even more confusing than Figure 3. While it is said at multiple points in the

text that it is described how this experiment is done, this is actually not true. The authors stop before the quantification and analysis. I can understand where the r-protein is coming from, but nowhere is it clear what the total protein is. Is this the sum of all r-proteins, as implied? If so, then this analysis is totally non-sensical. Because if all proteins go down as in panels C and D, then there should be straight lines, as the decrease would be normalized out. If some other measure of total protein is used, then this measure must be shown in a Figure. It could be a supplemental Figure. Along those lines, Figure S4 should show the raw data also for a depleted large subunit protein. Finally, while the analysis in Figure S4 is admirably quantified, it is only linear in a small range, and the measured protein concentrations fall out of this range, and can thus not be used. This is very clear in the standard curves, which have more data points than the curves (ie some were discarded), and the range is just 2 fold, and the observed depletions are more than that.

Furthermore, an important problem with quantitative Westerns is whether the ratio of two proteins remains constant over the range of concentrations used. Indeed Figure S4 shows that this is not the case. While the ratio of uS4 to uL18 might be the same at each concentration, the ratio of each of these proteins to uL4 is different at each concentration. This can be the case even if the standard curves are linear if the slopes and y-intercepts vary a lot, as they clearly do. Thus, this analysis, even though very thorough, is likely not valid.

6. Figure 5A suffers clearly from effects due to the switch in the carbon source. This is why ITS1 and ITS2 in BY cells increase. This observation is ignored, but will completely compound the analysis. The authors must repeat this experiment without a Gal switch as delineated above. In addition though, ITS1 levels are a measure of a lot of stuff: rates of transcription, rates of processing, rates of degradation; So, drawing any conclusion from these levels cannot be done without also addressing all of these.

7. Figure 5B needs several additions: First of all, both wt and uL4 depletion must be done and shown at 8h (or the wt control at 6 h). Most of the experiments in here were done at 8h past depletion, so this is the most important timepoint. Secondly, the experiment shows clear differences between the strains, which are ignored. In addition, this experiment needs replicates, and error bars to evaluate quantitatively. This will be discussed more in point 12 below.

8. Figure 5E shows again that there are clearly effects on transcription as 5S is increased, and this Figure needs also to be repeated without the glucose shift. Otherwise, effects on processing cannot be clearly deconvoluted. As it stands, the data show a delay in 20S rRNA appearance, and increased rRNA transcription.

9. Finally, Figure 5D should be compared to Figures 1K&L. They both essentially show the same thing, but I get the sense that they do not give the same answer, as the accumulation as measured by polysome profiles seems to be higher than as measured by 18S/25S ratio.

10. eS31 is a late-assembling protein, and only assembles in the cytoplasm. It can affect the export of uS5 only if there are feedback effects (because of the lack of ribosomes now no new ribosomal r-proteins are made and that affects uS5 export. This is a key critical problem with the entire analysis in here as assembly is probed so late. This is also likely the reason why all proteins behave the same.

11. Figure S2 needs a control for undepleted ribosomes for comparison and the Westerns over the polysomes (they stop in the 80S fraction). Also, this experiment should also be carried out after 60S block. This is a really important and informative experiment if done with the proper controls.

12. The work in here shows that disrupting 60S assembly has effects in 40S but not vice versa. This has been known and noted for a long time (I believe first in the Venema and Tollervey review from almost 20 years ago). What the authors want to claim is that this is from post-assembly turnover of 40S. While I think it is possible from the data that this is the case, this is certainly not the only thing. And to actually draw that conclusion, and describe the data correctly, the authors need to do a much more thorough and quantitative analysis. Because a lot of stuff happens: transcription is changed because of the switch in carbon source, and because of the ribosomal

stress. In addition, processing is clearly affected as shown in Figure 5, and in many previous papers. These all affect these ratios, and it is not quantified how. Thus, it is critical that these effects be deconvoluted (by getting rid of the carbon-source effects), and quantified, to be actually able to draw any conclusions.

13. Secondary effects are not all considered: growth stops after about 2h, the analysis are done after 8h. AT that point there are clearly no polysomes, and the proteins affected include ribosomal r-proteins, the most highly translated class of mRNAs. Thus, after ribosome assembly stops, because of downregulation of a specific protein, no new ribosomal proteins are made, which affects other things.

Minor points:

14. eS1 is not an early binding protein, it is middle, uS11 is an early binding protein and uS7 is middle.

15. P.3, second paragraph: "ribosomal proteins are added in waves, [...] binding of subsequent waves depends on proteins in the previous wave." We now know that this is untrue. Assembly can proceed in the absence of missing proteins. Williamson has shown this for the large subunit and Culver for the small subunit.

16. P.5, last sentence before the new section: "as shown below, [...]". This sentence must go away. It is a) not shown and even if so, then this sentence should wait until it is actually shown. Same for the last sentence before the new section on p.8.

17. P.6, last sentence before the new section. "[...], indicating that growth rate affects dynamics of 55S accumulation". A single timepoint (16h) is assessed. Thus, no conclusions about dynamics can be made. By definition.

18. The authors should make it a lot more clear that this is something that happens after VERY prolonged depletion of RPs. Most people do not deplete their ribosomal proteins for that long before they assay these effects. This should be mentioned repeatedly, and specifically addressed in the discussion.

19. Figure 2: both the left and middle columns have the same label. That is presumably incorrect. If it is correct, then there is a lot of variation between experiments..

20. Figure 4 needs x-axis labels.

21. Figure S4B needs to have the labels in the new nomenclature.

22. To refer to a different paper for the yeast strains in this manuscript is unacceptable.

23. The authors might want to consider doing their polysome analysis at lower Mg to reduce the 80S peak in favor of free 40S and 60S. This is described in Bhattacharya et al., MCB 2010.

January 16, 2019

Andrea Leibfried, PhD
Executive Editor
Life Science Alliance

Dear Dr. Leibfried,

We appreciate the suggestions and comments made by you and the reviewers. We have carefully studied them and revised the manuscript extensively, including adding extra data. We have of course also responded carefully to the reviewer's comments (in red below each of the referee's comments).

We further note that while we were working on our revisions, a paper scheduled for Molecular Cell in January 2019 has appeared online (<https://www.ncbi.nlm.nih.gov/pubmed/30503772>). It contains proteomic data supporting our conclusions that reducing expression of 60S r-protein genes affect accumulation of 40S, but not vice versa. However, their manuscript only addresses the equilibrium in steady-state growth, while we have studied the transition from balanced to imbalanced synthesis of r-protein genes. Furthermore, Cheng et al. does not include an in-depth analysis of ribosomal particles.

Reviewer #1:

In this manuscript the authors have used a number of biochemical approaches to address the question whether 40S and 60S assembly/accumulation of ribosomal subunits is interdependent. The results show that blocking 60S subunit synthesis impairs 40S accumulation and nuclear export, however, blocking 40S synthesis does not significantly impair 60S biosynthesis or export. The data indicate that this is independent of rRNA transcription and that the observed decrease in 40S subunits is most likely the result of degradation of excess 40S. **The idea that 40S and 60S assembly/accumulation is coupled (to some extent) is not novel**, however, this relationship has not been so carefully examined up to now. Overall the data support their conclusions but I have a few suggestions that may improve the manuscript.

We are pleased with the reviewer's concluding remark, but we are not aware that the issue of **accumulation and compensation for a distorted balance between the subunits** has been given any significant attention in past publications. The asymmetry between the effects of cessation of subunit biogenesis on the accumulation of the other subunit is novel. In fact, proteomic data leading to the same conclusion are published in Molecular Cell this month.

Comments:

Page 4: The authors discuss on page 4 that depletion of LSU proteins results in a decrease of 60S subunits but 40S levels were increased relative to other peaks. They conclude from this that 40S subunits were still produced. From these data alone I'm not sure I would draw this conclusion as there is no data showing the effect of 60S depletion on pre-40S rRNA processing. But later on page 9 they show nice data that suggest that 40S accumulation is in fact inhibited in response to blocking 60S biogenesis. For me this was a bit confusing. I am wondering if the authors could move Figure 4 into Figure 2 and I would also recommend adding Supplementary Figure 2 to the main figures as this is

really an excellent control experiment. I would even consider adding an extra figure showing (lack of) incorporation of a GFP-tagged SSU r- protein. I think moving the figures around might help to improve the flow of the manuscript as it is then immediately clear to the reader what the effect of the depletions have on 40S and 60S assembly. Then the authors could then discuss the 55S particles.

We have rearranged the manuscript to accommodate the reviewer's good suggestion.

Page 9: The following sentence is quite lengthy and a bit difficult to follow. I would recommend rephrasing this: "In contrast to the pan-subunit effect of repressing 60S protein synthesis, restraining the synthesis of the 40S proteins uS4, uS10, or uS11 generated a subunit- specific response, i.e. the specific concentration of uS4 protein declined, while the abundance of 60S proteins uL4, uL5, and uL18 changed little (Figure 4E-G)."

We have simplified the text.

Figure 5. The first two panels show graphs of ITS1 and ITS2 abundance, yet the header above the graphs says "ITS1 abundance". Should this not be "ITS abundance"?

Thank you for catching this. We have changed the header

If I would be really picky, I would argue that ITS1 abundance slightly decreases due to r-protein depletion. It is not statistically significant for most time-points but there is an effect. Perhaps the authors could state here that there is a mild effect on ITS1 accumulation but there clearly isn't any indication that rRNA transcription is significantly affected. The run- on experiment is important but the figure shows only a few slices of blots and there is no quantification/normalization. It would be very helpful if the authors could quantify the run-on results and provide error-bars.

We have changed the text to acknowledge the slightly lower levels of ITS1 and ITS2 (relative to total RNA) during inhibition of subunit assembly.

The image of the blot from the run-on experiment shows the full blot (now Figure 6E). We have added quantification of the blots. Furthermore, we agree that ideally statistics on the run-on experiment would be nice. However we have added results from a primer extension experiment showing that A2/A3 5' ends resulting from rRNA cleavage are found six hours after blocking uL4 synthesis. Since these ends are unstable (ITS1 is degraded during processing), the 5' A2/A3 ends could only exist if rRNA transcription continues. We therefore argue that the combination of **three** different types of experiments (ITS1/ITS2 measurements, run-on, and primer extension) all pointing to the same conclusion is more persuasive than repeating the same experiment. Thus, we feel comfortable concluding that blocking uL4 synthesis does not significantly reduce the rRNA transcription relative to the total RNA content ($d(\text{rRNA})/(\text{total RNA})$).

Figure 6. Here the authors have used GFP-fusion r-proteins to look at export of ribosomal subunits. Why did the authors look at cells after 16 hours of gene repression? This seems quite long and, judging from the vacuoles, the cells are on their way out.

Also, to be able to have a complete picture of the data, would it not be helpful to also show the localization of not repressed cells?

We have inserted a comment to acknowledge that the effect on 40S export during abolishment of 60S assembly could be a secondary effect. Nevertheless, as pointed out in the discussion, long-term, possibly secondary, effects are relevant to understanding phenomena such as ribosomopathies.

We have also inserted measurements of cell viability during depletion of uL22/L17 (new Figure 1B). The viability does not decrease during repression of 60S assembly.

Finally, just a warning, we have had serious problems in the lab with this r-protein GFP reporter system. We noticed that when we integrated the GFP into the genome downstream of r-protein genes we often got a completely different result. Secondly, we have had cases where the rpS-GFP reporter showed clear nuclear accumulation, whereas FISH data showed clear cytoplasmic accumulation of 20S. Unfortunately, negative data is very difficult to publish...

I obviously won't insist that the authors do these experiments for this manuscript because they may have had different (more positive) experiences, but in the future I would strongly recommend using FISH to get a read-out of nuclear export of pre-ribosomes.

We understand the reviewer's concern. However, we received these constructs from the Hurt lab; according to their publications no problems akin to the reviewer's observations were obvious.

Discussion page 13. "Since assembly of the two ribosomal subunits occurs along separate largely independent pathways (Woolford and Baserga 2013), we were surprised that inhibition of the 60S subunit prevents accumulation of the 40S subunit."

I do not find this very surprising as it has been shown in several cases that depletion of LSU assembly factors can lead to 18S pre-rRNA processing defects. Perhaps a classical example is Rrp5 and I believe some of the HEAT repeat proteins show the same phenotype. Also the co-transcriptional processing model is not compatible with the idea that the processes happen independently. For example, co-transcriptional processing at site A2 requires Pol I to transcribe at least parts of the 25S sequence. It has been estimated that ~70% of the pre-rRNA is co-transcriptionally processed. Perhaps this would be worth highlighting this in the discussion as well. It might also be worth discussing the half-lives of the individual subunits. Is the 40S generally more stable than 60S? Could this partially explain why after 8h of depletion so much 40S is still present?

We have removed "surprising" from the text and inserted a section in the Introduction to point out that depletion/inactivation of **some** factors affect the production of both subunits, while depletion of other factors specifically affect the formation of **only** one subunit. In agreement with this, our pulse chase experiment shows that 18S is fully matured (at least according to electrophoretic mobility) at least 6 hours after repression of uL4 synthesis. This is also evident from a number of papers reporting expression of other factors specific to the 60S maturation pathway (see references in the revised Introduction). Accordingly, we have modified the text to emphasize that the 40S and 60S assembly pathways are independent **after**, but **not before**, the separation of the 20S and 27S segments of the primary transcript. The view is also commensurate with the Table 3 and 4 in the review of Woolford and Baserga.

Reviewer #2 (Comments to the Authors (Required)):

The presented manuscript addresses the question of if and how cells ensure that equal levels of the small and large ribosomal subunits are produced in situation when ribosome biogenesis is disrupted. The authors describe an interesting observation that inhibition of either small (40S) or large (60S) ribosomal subunits synthesis pathways does not lead to the same outcome. While inhibition of the 60S synthesis reduced also accumulation of the mature 40S subunits in their experiments, the opposite was not true. Interestingly, authors' experiments suggest that 40S synthesis continues and reach perhaps partial maturity (at least based on presence of 18S rRNA), however, these subunits are perhaps not fully exported to cytoplasm. The data presented provide some insight into these intriguing questions. However, there is a number of issues in the current version that need to be answered.

Figure 1. A block in the 60S synthesis typically leads to the appearance of halfmer peaks in the polysomes (polysomes, where the 40S initiation complex is bound and waiting for the 60S to join). Could author provide on their explanation why it is not the case here (at least not at the 2.5h time-point in the Fig 1D)? Does the lack of halfmers indicate that the translation initiation was arrested?

Even a short incubation with cycloheximide before harvest induces halfmer formation and increases the polysomes relative to the total ribosome mass (Helser et al 2009). We interpret this to mean that cycloheximide affects the balance of protein synthesis initiation and termination and thus could be considered an artifact. Hence, we chose **not** to add cycloheximide to the culture prior to harvest, although we did add it to the lysis buffer to “freeze” polysomes post-lysis. We have added a comment to the Materials and Methods to emphasize that we did not add cycloheximide to cultures prior to harvest and why.

Also, it would be informative to show the growths curves during the depletion for all the relevant figures. Were the cells still growing at 8h or 16h time points? Does the reduction of the total ribosome content correspond to the "dilution" of ribosomes by cell division or does it indicate an active degradation/turnover process of the remaining ribosomal subunits (perhaps linked to the increased vacuoles)?

We have added growth curves for several strains and calculated dilution curves for repression of uS4 and uL4. In both cases the specific abundance (r-protein/lysate A280) follows kinetics expected based on the dilution curve. Thus there is no need to invoke active degradation. However, the vacuoles might participate in the “clean up” of abortive assembly intermediates, or potentially a massive rearrangement of the cell's protein composition. However, investigating turnover of ribosomal and other proteins is beyond the scope of this paper.

Figure 1. It is not clear if the presented polysome profile curves were normalized to the total A260 signal (clipping of the signal in the figure 1G and 1H indicates this was not done).

We have made clear that (i) constant A260 units were loaded on gradients that are to be compared and (ii) the colorimeter (254 nm) was set at the same sensitivity.

There is no visible green circle for the uL4 levels at 16h in the Figure 1K and 1L. Is it overlapping (then change the figure to show it) or missing?

The sucrose gradient for uL4 repression was done for 6 and 8 hours only. We have clarified the text to indicate that the figures showing 40S as a function of time are based on the collective data from repression of uL22, uL4, Rrs1, and Rpf2 (Figure 2M-N in the revised manuscript) and pointed out that the results from all the experiments together describe a single curve.

Figure 3. The panel A, methylene blue staining - the quality of the figure is not great. The 18S seems to present in all the fractions at time 0, which should not be the case. There should be at least 1 fraction gap (e.g. in this case no signal in fractions 13 and 14). This indicates trailing of the 40S/18S in the gradient (or mixing during the collection of fractions) and prevents any meaningful interpretation. In addition, the presence of strong degradation smear in the critical lanes 8 and 10 could easily obscure any weaker signal, again preventing solid conclusion whether any 18S is or not present in the 55S peak. Ideally the quality should be improved, or the membrane probed also with a radioactive 18S probe.

Figure 3B,C. Why is there no signal for 27S and 25S rRNAs for the time 0 lanes, especially in the lane 11 corresponding to 60S peak? Both 25S and 27S must be present at time 0. This pattern does not correspond at all to the methylene blue staining, where 25S rRNA is clearly present in all "0h" lanes.

The resolution in the gradients in the original Figure 3 were loaded with more material than the gradients in the original Figure 1 resulting in reduced resolution. We also overloaded the northern blots to optimize the view of 27S pre-rRNA. We have replaced the methylene blue stained blot in the original manuscript with blots from each of the gradients in a separate experiment so that the blot can more easily be correlated with the sucrose gradient traces. These blots also clearly show the fragmentation of rRNA in the 55-60S peaks in the 8 hour, but not the 0 hour, sample (also visible in the old figure, but it may be confusing with the alternating 0 and 8 hours sample).

We have re-optimized the contrast of the northern blots. The reviewer's concern about "no signal for 25S and 25S for the time 0 lanes" stems from the fact that the lanes of the 8 hour samples flanking the 0 hour sample contain a lot more 27S and 25S rRNA. Note that the slots were loaded with equal portions of each sucrose gradient fraction and the 60S peak is higher in the 8 hour sample. We think the view of individual lanes is now clearer in the adjusted image. To further clarify the content of rRNA in the critical lanes, we have electronically excised critical lanes from the full northern image and optimized contrast and brightness separately for each lane. This shows that there is a clear 25S band in the 60S peak at t=0 and that it is not significantly fragmented (the one breakdown product does not match the fragmentation pattern at t=8), and that there is no fragmentation of the 25S rRNA in the 80S peak at either of the two time points.

Finally, we have deleted the discussion of 18S in the 55S peak and emphasized that the distributions of uS4 and 60S proteins in the western blot clearly indicate the lack of 40S material in the 55S peak. Note that uS4 is a key protein for the initial folding of small subunit rRNA (papers by the Draper and Woodson labs), so a pre-40S could not form in the absence of this protein

Figure 3D and 3E. and the last sentence of the second paragraph, page 8: "Thus, the fragmentation must involve endonucleolytic attacks". This is an important experiment, but the conclusion is not fully correct. The 5'- and 3'-end probes label a different pattern of bands. Such pattern can arise also from a mixture of 5'>3' and 3'>5' exonucleases and strictly speaking does not have to be made by an endonuclease. Please correct the statement

Good point, thank you. We have corrected the text.

Figure 4. The authors show behavior of few other L proteins during L protein depletion, indicating coregulation/co- degradation of L proteins (as expected). Could they also show the behavior of other S proteins during an S protein depletion? Are also S proteins coregulated.

The referee is making a good suggestion, but we do not have antisera for other 40S proteins.

Also, the charts miss the X-axis labelling.

Thanks for pointing this out. We have corrected the problem.

Figure 5A-C,E. Pre-rRNA abundance. Authors claim that pre-rRNA abundance is increased during depletion. However, the experimental setup does not necessarily support this conclusion. Authors used a slot blot, where same amount of total RNA was deposited in each slot. There are two major issues with this:

The total RNA in later time points contains significantly less of mature ribosomal RNA (which in growing yeast cells represents 80% of total RNA). In the later time points of the depletion, when the total ribosome levels are strongly reduced, the total RNA will in proportion contain much more of non-mature rRNAs, including more of the pre-rRNAs. It thus leads to overloading of the remaining RNAs and artificially increasing the amount of precursors, obscuring the well-known and established reduction of the rRNA transcription per cell. These experiments need to be done by loading the same number of cells. Otherwise, only a qualitative statement that some (most likely strongly reduced) transcription continues can be made. The same applies to the run-on experiments and pulse-chase when same amount of total RNA is loaded.

We apologize for using the term "rate" in the original submission. We should have referred to the "specific rate", i.e. $d(\text{rRNA}/dt)/\text{total RNA}$. Note that we are normalizing to A260 of the lysate (interpreted as **total RNA**), not total **rRNA**. We have inserted a figure showing that the yield of A260 material per OD600 unit harvested does not decrease (Figure 6F in the revised manuscript). While we have not investigated the background for this somewhat surprising result (and think that such investigation is beyond the scope of the current manuscript), we think that the shift-up effect stimulates the synthesis of other types of RNA (mainly tRNA) to compensate for the loss of ribosome accumulation. This would agree with the fact that A260/OD600 increases after the shift in the control strain, but not in the strain where uL22 synthesis is repressed.

Regarding the pulse-chase experiment (Figure 6G) in the revised manuscript) it can only be interpreted to answer whether 18S rRNA is processed. As we have explained in the

revised manuscript that the incorporation of radioactive uracil is reduced, because much of the transcribed rRNA is degraded during inhibition of uL4 synthesis. Thus the **net** drain on the triphosphate pools is reduced, resulting in a slower rise of the specific activity of the pyrimidine triphosphate pools. The pulse-chase experiment does therefore not generate information about the rate of rRNA synthesis, only the efficiency with which the transcribed is processed into mature rRNA. Since all lanes are loaded with the same amount of radioactivity (30,000 cpm) and incorporation is stopped at time 0 due to the addition of “cold uracil”, the efficiency of 18S processing at 0 and 6 hours can be compared from the strength of the bands in the 20-minute chase.

Finally, please note that all our arguments are based on the specific rRNA synthesis ($d(\text{rRNA})/d(\text{total RNA})$) and the relative composition of ribosome fractions. (i) Does the specific rRNA synthesis change? Not much. (ii) Does the free 40S/total ribosomes increase perpetually? No it does not. (iii) Do the 40S and 60S r-proteins co-vary? Yes for repression of 60S assembly, no for repression of assembly of 40S subunits. (iv) Is the efficiency of 18S processing decreased? No.

Incidentally, question (iii) is the same asked in the global proteomic analysis of steady-state-growing mutants lacking one of the paralogous r-protein genes (Cheng et al. *Mol Cell* **73**: 1-12). These authors concur with our conclusion (see also comment above).

Loading total RNA in a slot blot also prevents distinguishing between different pre-cursors. The ITS1 and ITS2 probes are complementary to several different precursors, such as the primary transcript 35S pre-rRNA (which will be detected by both probes!). It is well established that the 35S levels increase during aberrant ribosome biogenesis, but not necessarily due to more transcription but due to stabilization/arrest of processing. In addition, both probes will also detect potential contaminating genomic DNA (which contains 200 repeats of rDNA) contamination. Therefore results from the slot blot these results are inconclusive.

We used slot blots precisely to conveniently measure the **sum** of all forms of ITS1- or ITS2-containing precursors. Note that ITS1 and ITS2 were measured separately. The results are essentially the same for the control strain (BY4741) and the Pgal-ul4 and -uS4 strains. Since the transcribed spacers are unstable, this indicates that rRNA transcription continues for at least 8 hours. If transcription had stopped the content of transcribed spacer would have declined significantly, potentially to zero. See also our answer to the previous comment regarding normalization to A260 (total RNA).

As to the potential error from DNA contamination in the RNA preparations, we note that comparing the rDNA copy number with the amount of precursor rRNA shows that such contamination would be insignificant: There are ~150-200 rDNA repeats in the genome (Dammann et al 1993), i.e. at most 400 copies per cell just before cell division. However, the cells make about 1000 ribosomes per minute, since the cells contain ~200,000 ribosomes (von der Haar 2008; Warner, 1999) and the doubling time is ~150 minutes. Removal of ITS from the rRNA transcripts during processing takes 5-10 minutes, i.e. the life time of the transcribed spacer is 5-10 minutes (look at the appearance of 5.8S rRNA and the disappearance of 27S and 20S in the pulse-chase experiment, which closely resembles pulse-chase experiments in several other papers). Thus the cell contains in the order of 5,000-10,000 precursor molecules harboring ITS1 and ITS2 compared to the 150-300 copies of rRNA genes making any rDNA contamination in the RNA preparations irrelevant. We therefore disagree with the referee's conclusion regarding the slot blots.

Figure 3E. In the discussion section authors refer to this figure to show presence of 23S pre-rRNA. However, there is no label for the 23S band. While I can see some smear/weak bands in that are, it is not clear to which authors refer. Please include an arrow for the 23S band if clearly present and if not, correct the statement in the discussion.

Apologies. The important short word “not” was inadvertently left out, in spite of intensive proof reading.

Reviewer #3 (Comments to the Authors (Required)):

In this manuscript Gregory et al use a combination of Northern and Western analyses and immunofluorescence to study what happens to both ribosomal subunits when assembly of one or the other is blocked for extended times.

The reviewer’s comment “extended timed” seems to focus only on the 8-16 hour time points. Our time points for sucrose gradients begin at 2-2.5 hours, while our analysis of ribosomal components begin at 1 hour. We disagree with describing this as “extended times”. Furthermore, as emphasized in the Discussion, the long-term effects of the distortion of ribosome biogenesis are relevant to phenomena such as ribosome-related diseases (ribosomopathies).

This manuscript recapitulates observations known for a long time in the literature and combines them with some more in depth analysis. Overall this manuscript requires a more consistent analysis, overstates much of the data and is also incorrect in several instances.

Details are below.

Major points:

Given that none of the observations in this manuscript are new, the strength would lie in a careful analysis.

For starters, the authors set themselves up poorly by switching everything from galactose to glucose. This switch from a so-so nutrient source to their favorite food affects ribosome assembly, and the analysis herein, but is not considered anywhere. A better way to do this would have been to use the strains they use, and then either have or not have plasmids encoding the ribosomal proteins under constitutive promoters, but analyze everything in glucose. While it might not be necessary to repeat all of the analyses in here, a subset of the experiments needs to be repeated using this cleaner system to deconvolute effects. This is minimally true for the transcriptional analysis, and the analysis of rRNA levels.

First, it does not matter how the balanced production of the ribosomal subunits is perturbed as long as it is balanced in the start condition. In our original submission we left out most of the sucrose gradients for $t=0$, but we have now added 0-time gradients. By comparison to numerous other papers, they look like sucrose gradients of lysates from uninhibited growth of standard yeast strains, documenting the balanced production of subunits.

Second, we did the same gal-glu shift experiments with the BY4741 control strain that has a

normal complement of chromosomal r-protein genes. Any deviation from the results obtained with this control strain can be ascribed to the repression of the r-protein genes under gal control. Third, any other perturbation of the assembly, such as controlling the expression of r-protein genes from a tetracycline promoter, all have their secondary effects. In all cases, the interpretation depends on comparing with the parent strain as we have done. Fourth, the experiment proposed by the reviewer cannot be done and would be uninformative if it could be done. The proposed strains **without** a complementing plasmid are identical to the strains we actually used, and they do not grow in glucose as explained in the manuscript. Furthermore, inserting a plasmid with the relevant r-protein gene expressed from a constitutive promoter would complement the repression of the gal-controlled gene and result in uninterrupted synthesis of all r-proteins and unperturbed ribosome formation. The plasmid would simply result in permanent over production of the protein whose gene is carried by the plasmid and this excess would be rapidly degraded as shown many years ago by the labs of Warner, Friesen, and Woolford. That is, the strain would recapitulate “wildtype” growth. Thus the experiment proposed by Reviewer 3 would not provide any information for answering the question we address: Are there mechanisms that rectify unbalanced formation of the two subunits? We could have used strains with deletions of one of the paralogous gene where a protein is encoded by two genes. This is essentially the approach taken by Cheng et al, but their experiment only allows conclusions of the final equilibrium state, not the transition of uninhibited ribosome biogenesis to blocking assembly of one of the subunits. Our experiments address the transition.

Figure 1, Figure2, Figure S1 and Figure S3 show sucrose gradients for depletion of various small and large subunit proteins. But there is no consistency. For two proteins an entire time course is shown, but the two time courses are different.

Time courses are shown for repression of **three** r-protein genes; uL22, uS4, and eS31. Since the outcome of the experiment differs depending on whether a 40S or a 60S protein gene is repressed, the time courses are designed differently to highlight the separate outcomes.

Then, for many the t=0 time point is missing. This is critical, because Gal overexpression of some proteins is actually detrimental. The authors should provide time courses, including minimally a 0, 6 or 8h and 16h timepoint for all RPs.

We originally left out the t=0 gradients because they all look the same and we wanted to save space. We have now inserted these gradient tracks (we have removed the eS1 experiment, since we apparently did not save the t=0 trace). All gradients look like typical sucrose gradients of cells growing unperturbed ribosome synthesis.

As far as time courses, we have shown sucrose gradients for the same 6 time points for an early (uS4) and a late (eS31, in the supplement) 40S assembly protein. For the 60S assembly, we have only shown a full time course for uL22. However, a compilation of time points for the abundance of 40S subunits after repression of two 60S r-proteins and two 60S assembly factors all fall on a single curve, verifying that there is no substantial difference in the kinetics of the changing ribosome composition. Furthermore, we have shown the abundance of ribosomal proteins and rRNA ratio after repressing three different 40S and three different 60S r-protein genes for 1, 2, 4, and 8 hours for. We think this is sufficient to show the principle effects of abrogating assembly of each of the subunits: Repression of 60S r-protein genes abolishes accumulation of both 60S and 40S subunits,

while repression of 40S r-protein genes inhibit accumulation of only 40S subunits.

In Figure 2A&B, they want to draw conclusions about the 55S peak not having uS4, when uS4 is depleted. This is clearly not a valid conclusion. In addition, there is concern that in panel C, uS4 is not found in the 40S peak.

As summarized in the manuscript text, the distribution of r-proteins in panel A (t=0) is as expected. In panel B (t=8), the 60S proteins (uL4, uL18 and uL5) are all present in 55S, 60S 80S and polysomes. However, uS4 are only found in 80S and polysomes as is evident from comparing the distribution of uS4 with the distribution of the 60S proteins (uS4 is only found in fractions sedimenting $\geq 80S$ at t=8). Thus we think it is valid to conclude that 55S does not contain uS4 and must respectfully disagree with the reviewer. In panel C no uS4 is found in the 40S peak, because the figure shows the 8 hour time point for depletion of uS7 and there is essentially no free 40S left.

Figure 3 is very confusing. Why is there essentially no signal for the 25S 5' in the 0h depletion? This probe should pick up mature 25S, of which there is at least as much in the undepleted cells.

See answer to comment of Reviewer 2 on the same point.

Also, panels D and E require a control for undepleted sample.

We respectfully disagree. The 25S in the 60S peak at t=0 is not degraded (Figure 8D-E in the revised manuscript, especially the enhances lanes in panel E). The important question is to compare the 5' and 3' probes at t=8 when the fragmentation is evident.

Furthermore, this Figure shows that their gradients are not able to effectively separate anything from anything as both 18S and 25S are all over the gradient. This must be repeated in a cleaner way, as it also casts serious doubt on what the 55S complex actually is.

See answer to comment of Reviewer 2 on the same point.

Finally, the authors claim that Figure3 shows that there is no 18S rRNA in the 55S peak. It is totally unclear how one could make such a claim. These lanes are so smeary from the degradation products observed that one would never be able to see 18S rRNA. If the authors want to make such a claim, they must do it by Northern probing, with at least 2 probes at different locations to rule out degradation of 18S.

See answer to comment of Reviewer 2 on the same point.

Figure 4 is potentially even more confusing than Figure 3. While it is said at multiple points in the text that it is described how this experiment is done, this is actually not true. The authors

stop before the quantification and analysis. I can understand where the r-protein is coming from, but nowhere is it clear what the total protein is. Is this the sum of all r- proteins, as implied? If so, then this analysis is totally non-sensical. Because if all proteins go down as in panels C and D, then there should be straight lines, as the decrease would be normalized out. If some other measure of total protein is used, then this measure must be shown in a Figure. It could be a supplemental Figure. Along those lines, Figure S4 should show the raw data also for a depleted large subunit protein. Finally, while the analysis in Figure S4 is admirably quantified, it is only linear in a small range, and the measured protein concentrations fall out of this range, and can thus not be used. This is very clear in the standard curves, which have more data points than the curves (ie some were discarded), and the range is just 2 fold, and the observed depletions are more than that. Furthermore, an important problem with quantitative Westerns is whether the ratio of two proteins remains constant over the range of concentrations used. Indeed Figure S4 shows that this is not the case. While the ratio of uS4 to uL18 might be the same at each concentration, the ratio of each of these proteins to uL4 is different at each concentration. This can be the case even if the standard curves are linear if the slopes and y-intercepts vary a lot, as they clearly do. Thus, this analysis, even though very thorough, is likely not valid.

We apologize, but there was an error in the original manuscript: the standard curves were not used to normalize the r-protein abundance for the reason mentioned by the reviewer.

Changes in the manuscript made in response to this comment of Reviewers 1 and 3:

(i) The data in the old Figure 4 is now presented separately for repression of the 60S and 40S assembly (Figures 3 and 5, respectively).

(ii) We have expanded the description of the procedure for western quantification of r-proteins, which should clarify the experimental procedures. It should now be clear that each lane was loaded with the same number of A280 units of the extract and the bands are thus normalized to A280 loaded (left side of Figures 3 and 5).

(iii) We have also normalized the data for uS4, uL18, and uL5 to the data for uL4, essentially using uL4 as an internal standard. This shows that with the exception of uL18 after repressing uL4 synthesis (discussed separately in the Discussion Section), the 40S protein uS4 co-varies with the 60S r-proteins after abrogating 60S assembly, but **not** during abolition of 40S assembly.

(iv) Tables of raw data for protein measurements have been added to the Supplement

Figure 5A suffers clearly from effects due to the switch in the carbon source. This is why ITS1 and ITS2 in BY cells increase. This observation is ignored, but will completely compound the analysis. The authors must repeat this experiment without a Gal switch as delineated above. In addition though, ITS1 levels are a measure of a lot of stuff: rates of transcription, rates of processing, rates of degradation; So, drawing any conclusion from these levels cannot be done without also addressing all of these.

We explained above why the experiment suggested by the reviewer **does not work**.

We have now discussed in more detail the effect of the shift-up. As pointed out above, BY4741 was used as control and any deviations between BY4741 and the depletion strains can be ascribed to the disruption of ribosomal subunit assembly.

Figure 5B needs several additions: First of all, both wt and uL4 depletion must be done and shown at 8h (or the wt control at 6 h). Most of the experiments in here were done at 8h past depletion, so this is the most important timepoint. Secondly, the

experiment shows clear differences between the strains, which are ignored. In addition, this experiment needs replicates, and error bars to evaluate quantitatively. This will be discussed more in point 12 below.

This comment must refer to Figure 5C, not 5B. We disagree: question addressed by this experiment is whether rRNA transcription continues after the shift. We have added quantification of the bands in the run-on experiment. They confirm that transcription continues after the shift, supporting the conclusion from the measurements of ITS1 and ITS2. See also our answers to Reviewer 1 on the same issue.

3. Figure 5E shows again that there are clearly effects on transcription as 5S is increased, and this Figure needs also to be repeated without the glucose shift. Otherwise, effects on processing cannot be clearly deconvoluted. As it stands, the data show a delay in 20S rRNA appearance, and increased rRNA transcription.

We have commented above on the reviewers suggested experiment and why it does not work and the fact that shift-up effects are evaluated by comparing to the galactose culture. We have added text to the manuscript emphasizing that this experiment is not measuring rate of rRNA transcription, but efficiency of 18S processing. See also comments made to Reviewer 2 above.

We have discussed the delay in the appearance of 20S. Publications by the Tollervey and Woolford labs show that this is due to a shift from co-transcriptional to post-transcriptional rRNA processing (Axt et al; Talkish et al). However, this does not change the efficiency of 18S processing. See also comments above.

Finally, Figure 5D should be compared to Figures 1K&L. They both essentially show the same thing, but I get the sense that they do not give the same answer, as the accumulation as measured by polysome profiles seems to be higher than as measured by 18S/25S ratio.

The total 40S/total ribosomes increase by $(0.5/0.3)-1=66\%$ when considering all points from depletion of uL4, uL22, Rrs1, and Rpf2. The 18S/25S increases 40-60% after repressing the genes for uL18 and uL4. The 18S/25S ratio after repressing uL40 (a very late assembly protein) synthesis goes up 30%, then returns to 1 after some time. The difference between 40-60% and 66% is hardly alarming. In fact, **all experiments support our general conclusion**: repression of 60S assembly affects both 60S and 40S, but abrogation of 40S assembly has no effect on 60S assembly (at least not in the first 4-6 hours). The difference in the 18S/25S kinetics after repressing uL4 and uL18 vs eL40 may lie in the turnover rates of partly assembled precursor ribosomes that cannot be fully assembled due to the lack of proteins. However, investigating this would be beyond the scope of this manuscript.

eS31 is a late-assembling protein, and only assembles in the cytoplasm. It can affect the export of uS5 only if there are feedback effects (because of the lack of ribosomes now no new ribosomal r-proteins are made and that affects uS5 export. This is a key critical problem with the entire analysis in here as assembly is probed so late. This is also likely the reason why all proteins behave the same.

Contrary to the reviewer's statement, there is evidence that eS31 is added in the nucleus (Fernandez-Pevada et al NAR 44:7777 (2016)). Furthermore, the reviewer's suggestion would predict that repression of 40S proteins should also affect 60S assembly and nuclear export. This is not the case. Repression of 60S protein genes block accumulation of 40S subunits, but the opposite is not true.

Figure S2 needs a control for undepleted ribosomes for comparison and the Westerns over the polysomes (they stop in the 80S fraction). Also, this experiment should also be carried out after 60S block. This is a really important and informative experiment if done with the proper controls.

We agree that the control suggested by the reviewer would have been nice. However, since the experiment is only confirmatory of the aggregate conclusions from the sucrose gradient analysis, abundance of r-proteins and 18S/25S ratios, we do not think that it is necessary to elaborate further on this experiment. We have however, expanded the description of the experiment and moved the figure into one of the main figures.

The work in here shows that disrupting 60S assembly has effects in 40S but not vice versa. This has been known and noted for a long time (I believe first in the Venema and Tollervey review from almost 20 years ago). What the authors want to claim is that this is from post-assembly turnover of 40S. While I think it is possible from the data that this is the case, this is certainly not the only thing. And to actually draw that conclusion, and describe the data correctly, the authors need to do a much more thorough and quantitative analysis. Because a lot of stuff happens: transcription is changed because of the switch in carbon source, and because of the ribosomal stress. In addition, processing is clearly affected as shown in Figure 5, and in many previous papers. These all affect these ratios, and it is not quantified how. Thus, it is critical that these effects be deconvoluted (by getting rid of the carbon-source effects), and quantified, to be actually able to draw any conclusions.

The old experiments have been reinterpreted in the light of a better understanding of pre- and post-transcriptional rRNA processing as indicated above. This explains the delay in appearance of 20S and increase in 32-35S rRNA (see comment above) and that these kinetic changes do not affect the efficiency of 18S processing.

We are not aware that the asymmetry between repressing 40S and 60S assembly has been shown previously, except for the Cheng et al paper in this month's Mol Cell; clearly, we did not know about the experiments by Cheng et al when we submitted our manuscript; see also comment at the top of our rebuttal.

Furthermore, we think that the conclusion that the effect on 40S accumulation by repression of 60S proteins is post assembly is well documented by the observations that (i) processing of 18S is completed (at least to the size of 18S rRNA, leaving open the possibility of incomplete modification of rRNA and/or r-protein as discussed in our manuscript), and (ii) repression of early and late assembly proteins have the same effect; thus repression of the synthesis 60S proteins incorporated late, i.e. after A2/A3 cleavage and separation of the 60S and 40S pathways, block 40S accumulation.

Secondary effects are not all considered: growth stops after about 2h, the analysis are

done after 8h. At that point there are clearly no polysomes, and the proteins affected include ribosomal r-proteins, the most highly translated class of mRNAs. Thus, after ribosome assembly stops, because of downregulation of a specific protein, no new ribosomal proteins are made, which affects other things.

Growth (as measured by culture optical density) slows after 2 hours, but does not stop (see growth curve now added to the manuscript). This is in agreement with a number of other publications using the same or similar strains (for the purpose of investigating the mechanism of ribosome assembly, not the kinetics of ribosomal subunit formation). It also makes sense, because ribosomes present at the time of the shift continue to make protein, even though the production of one or both subunits stops.

Regarding secondary effects: We actually **did** comment on secondary effects in the discussion: the 55S is not visible until 4-5 hours, raising the possibility that a nuclease has to be induced to produce the 25S rRNA fragmentation and conversion of 60S to 55S. And as commented above, secondary effects are interesting and important.

Minor points: eS1 is not an early binding protein, it is middle, uS11 is an early binding protein and uS7 is middle.

Thank you for the correction. Text is updated accordingly

P.3, second paragraph: "ribosomal proteins are added in waves, [...] binding of subsequent waves depends on proteins in the previous wave." We now know that this is untrue. Assembly can proceed in the absence of missing proteins. Williamson has shown this for the large subunit and Culver for the small subunit.

In our original submission we did point out that the hierarchy can be modified by growth conditions and referenced Talkish et al. We have now added further comments on the flexibility of the order of addition of r-proteins during manipulations of r-protein synthesis. Despite these deviations from a strict hierarchy, the general concept of sequential association of r-proteins with nascent ribosomal subunits is still accepted and the terms "early", "middle" and "late" binding proteins are still meaningful.

P.5, last sentence before the new section: "as shown below, [...]". This sentence must go away. It is a) not shown and even if so, then this sentence should wait until it is actually shown. Same for the last sentence before the new section on p.8.

The manuscript is extensively rewritten

P.6, last sentence before the new section. "[...], indicating that growth rate affects dynamics of 55S accumulation". A single timepoint (16h) is assessed. Thus, no conclusions about dynamics can be made. By definition.

We used the term "dynamics" as referring to a change between the formation and degradation of the 55S. We think this makes sense, but since we agree that it can cause confusion the term is now eliminated from our manuscript.

The authors should make it a lot more clear that this is something that happens after VERY prolonged depletion of RPs. Most people do not deplete their ribosomal proteins for that long before they assay these effects. This should be mentioned repeatedly, and specifically addressed in the discussion.

The reviewer ignores the fact that time points for sucrose gradients after repression of uS4, uS31, and uL22 begin at 2-2.5 hours. Moreover, for measurements of rRNA and r-proteins begin after 1 hour. We also clearly pointed out that some of the effects (plateauing of 40S accumulation after inhibition of 60S assembly and appearance of 55S after cessation of 40S assembly) are not seen until a few hours after repression. The possibility that these effects therefore are secondary to the original insult on ribosome assembly is mentioned in the Discussion. We also discuss that late time points are relevant to non-conditional mutations, such as those that cause ribosomopathies.

Figure 2: both the left and middle columns have the same label. That is presumably incorrect. If it is correct, then there is a lot of variation between experiments.

The labels are correct. The left column shows depletion of two different 40S proteins for 8 hours, including the distribution of r-proteins between the different ribosomal fractions. The middle column shows a time course for the changing distribution of ribosomal particles after repressing the uS4 gene. The gradient for the 8 hour point (panel I) is in complete agreement with the 8 hour gradient shown in panes B.

Figure 4 needs x-axis labels.

Thank you for pointing this out. The label has been added

Figure S4B needs to have the labels in the new nomenclature.

Labels have been added

To refer to a different paper for the yeast strains in this manuscript is unacceptable.

We have included a table of strains in the supplementary material

The authors might want to consider doing their polysome analysis at lower Mg to reduce the 80S peak in favor of free 40S and 60S. This is described in Bhattacharya et al., MCB 2010.

Dissociating 80S and polysomes to free 40S and 60S would add equal numbers of subunits to the free subunit peaks, thereby making the peaks approach molarity. This would work obscure the effects we have demonstrated by quantifying a separate fraction of ribosomes: the free 40S.

February 5, 2019

RE: Life Science Alliance Manuscript #LSA-2018-00150-TR

Dr. Lasse Lindahl
University of Maryland, Baltimore County
Biological Sciences
1000 Hilltop Circle
Baltimore 21250

Dear Dr. Lindahl,

Thank you for submitting your revised manuscript entitled "The small and large ribosomal subunits depend on each other for stability and accumulation". As you will see, reviewer #2 and #3 now support publication, while a few issues still need to get addressed (reviewer #3). We would thus like to invite you to provide a final version, addressing the remaining concerns of reviewer #3. Additionally, the following final revisions are necessary to meet our formatting guidelines:

- Please mention in the materials & methods or individual figure legends the number of replicates performed
- Please add callouts to Fig 3E, 4 A, 4J, 4K in the manuscript text
- Please add callouts to all panels of Fig S2, S3, and S4 in the manuscript text
- Please mention the individual panels in the legend for Fig S4
- Please upload Figure S3H-O as 'Source Data' for Figure S3

A. FINAL FILES:

-- High-resolution figure, supplementary figure and video files uploaded as individual files: See our detailed guidelines for preparing your production-ready images, <http://life-science-alliance.org/authorguide>

-- Summary blurb (enter in submission system): A short text summarizing in a single sentence the study (max. 200 characters including spaces). This text is used in conjunction with the titles of papers, hence should be informative and complementary to the title. It should describe the context and significance of the findings for a general readership; it should be written in the present tense

and refer to the work in the third person. Author names should not be mentioned.

B. MANUSCRIPT ORGANIZATION AND FORMATTING:

Full guidelines are available on our Instructions for Authors page, <http://life-science-alliance.org/authorguide>

Sincerely,

Andrea Leibfried, PhD
Executive Editor
Life Science Alliance
Meyerohofstr. 1
69117 Heidelberg, Germany
t +49 6221 8891 502
e a.leibfried@life-science-alliance.org
www.life-science-alliance.org

Reviewer #1 (Comments to the Authors (Required)):

The authors have made a significant number of revisions to the manuscript and have addressed all my concerns adequately. I would recommend publication.

Reviewer #2 (Comments to the Authors (Required)):

In the revised manuscript the authors satisfied all the comments and questions of this reviewer. I recommend accepting the manuscript.

Reviewer #3 (Comments to the Authors (Required)):

This revised version only partially addresses my previous concerns. Significant issues that remain are as follows:

1. p.15: "Western analysis of sucrose gradient fractions showed that the 55S peak contains large subunit proteins uL4, uL5, and uL18, but not the 40S protein uS4 (Fig 4BC), suggesting that this novel peak is related to the 60S subunit."

This was my original comment. The 55S is only observed upon depletion of uS4 (they didnt test another 40S protein). If that peak does not have uS4, bc there is no free uS4, then we cannot draw any conclusions about whether this peak is a 40S or 60S-related molecule. There are many other Abs available other than against uS4. This has nothing to do with anything about gradients at t=0. It is simple logic. No conclusion can be drawn about the 55S molecule.

Similarly, the explanation for why there is no 25S in the gradient Northernblots that is provided to reviewer 2 and 3 askign about it makes no sense. The authors are saying that they loaded equal volumes, and that the peaks increase at 8h. But they don't increase more than 2fold. Even if they increased 10 fold, Northernblots are linear in a large range...this should be visible. This Northern blot needs to be redone, or taken out of the manuscript and the conclusions must be modified.

2. While some ribosome biogenesis factors, such as Rrn5 and RNase MRP, are important for production of both ribosomal subunits (Lindahl et al. 2009; Lebaron et al., 2013), other factors, such as Drs1, Fal1, Has1, Rcl1, and Rrp1 work in only one of the two assembly pathways (Kressler et al. 1997; Billy et al. 2000; Emery et al. 2004; Horsey et al. 2004; Horn et al. 2011; Woolford and Baserga 2013; Talkish et al. 2016).

This sentence is wrong. Rrp5, Has1 and Prp43 are the three factors required for assembly of both subunits. It is unclear where the choice of the others is coming from...I would probably just state the first, not the second.

3. Mutations in genes for r-proteins and subunit-specific factors can distort the normal 1:1 production of the two ribosomal subunits, thereby affecting the translation process. For example, an excess of 40S subunits may sequester mRNA in initiation complexes that cannot be converted to translating ribosome because of the shortage of 60S subunits.

Each of these two sentences need a reference.

February 21, 2019

Dear Dr. Liebfried,

Thank you for the comments on our revised manuscript provided by you and the reviewers. We are pleased that you have invited us to submit the final version.

We have reacted to the comments by Reviewer 3 by clarifying sections of the text and written detailed rebuttals to the comments we disagree with.

We have of course also complied with your requests.

Sincerely,

Lasse Lindahl

Reviewer #1 (Comments to the Authors (Required)):

The authors have made a significant number of revisions to the manuscript and have addressed all my concerns adequately. I would recommend publication.

Reviewer #2 (Comments to the Authors (Required)):

In the revised manuscript the authors satisfied all the comments and questions of this reviewer. I recommend accepting the manuscript.

Reviewer #3 (Comments to the Authors (Required)):

This revised version only partially addresses my previous concerns. Significant issues that remain are as follows:

1. p. 15: "Western analysis of sucrose gradient fractions showed that the 55S peak contains large subunit proteins uL4, uL5, and uL18, but not the 40S protein uS4 (Fig 4BC), suggesting that this novel peak is related to the 60S subunit."

This was my original comment. The 55S is only observed upon depletion of uS4 (they didnt test another 40S protein). If that peak does not have uS4, bc there is no free uS4, then we cannot draw any conclusions about whether this peak is a 40S or 60S-related molecule. There are many other Abs available other than against uS4. This has nothing to do with anything about gradients at t=0. It is simple logic. No conclusion can be drawn about the 55S molecule.

As is clear from the headings in Figure 4C, this figure shows a gradient and western after depletion of uS7/S5, NOT depletion of uS4. Thus, the 55S peak appears after repression of any of the 40S genes tested, as well as after

repressing the 40S-specific assembly factor Rrp7. Furthermore, the distribution of r-proteins across the gradient **is the same** after repression of two different r-proteins, uL4 and uS7. That is, **uS4 is absent from the 55S peak when the uS7 gene is repressed, while the uS4 gene is not repressed.** Therefore, the absence of uS4 in the 55S peak cannot be due to depletion of uS4.

We do however understand that the text on page 15 of the manuscript can be confusing, because this text describes RNA analysis of the 55S peak after repression of uS4. We have now clarified our point by referring to Fig 4C in which uS7 synthesis is disrupted, but uS4 is expressed. We think it is a legitimate comparison, since the sucrose gradients, including the western analysis of gradient fractions, look the same after repression of uS4 and uS7 synthesis.

Similarly, the explanation for why there is no 25S in the gradient Northernblots that is provided to reviewer 2 and 3 askign about it makes no sense. The authors are saying that they loaded equal volumes, and that the peaks increase at 8h. But they don't increase more than 2fold. Even if they increased 10 fold, Northernblots are linear in a large range...this should be visible. This Northernblot needs to be redone, or taken out of the manuscript and the conclusions must be modified.

The reviewer argues that there is no evidence for 25S in the 60S during uninterrupted growth. We respectfully disagree with the reviewer's persistent concerns. Our arguments are as follows: (i) In the revised manuscript we replaced the methylene stained blot in the original submission with separate stained blots of the 0 and 8 hour gradients from a separate experiment. These blots clearly show intact 25S rRNA in the 60S peak (new Figure 8A-B) at 0 hours. (ii) The contrast of the northern blot was reoptimized (Figure 8D). It is now clear that the 60S peak at 0 hours contains intact 25S rRNA. (iii) We have electronically excised lanes from the 60S peak and optimized them individually for better viewing. The excised lane from the 0 hour 60S lane shows even clearer that there is intact 25S rRNA in the 60S peak at 0 hours, but degraded 25S rRNA at 8 hours. The excised lanes also show that the 25S in the 60S at 0 hours is **not** spill over from neighboring lanes, since there the degradation products seen in 60S at 8 hours are not seen in the 60S at 0 hours. Together these results clearly show that the 0 hour 60S peak contains intact 25S rRNA, while the 8 hour 60S lane contains degraded rRNA. The northern of RNA from a separate experiment confirms the degradation of 25S rRNA in the 8 hour 60S peak.

2. While some ribosome biogenesis factors, such as Rrn5 and RNase MRP, are important for production of both ribosomal subunits (Lindahl et al. 2009; Lebaron et al., 2013), other factors, such as Drs1, Fal1, Has1, Rcl1, and Rrp1 work in only one of the two assembly pathways (Kressler et al. 1997; Billy et al. 2000; Emery et al. 2004; Horsey et al. 2004; Horn et al. 2011; Woolford and Baserga 2013; Talkish et al. 2016).

This sentence is wrong. Rrp5, Has1 and Prp43 are the three factors required for assembly of both subunits. It is unclear where the choice of the others is coming from...I would probably just state the first, not the second.

Actually, there are more factors that are required for assembly of both

subunits (Klinge and Woolford, 2018). But to the point of the reviewer: The sentence is **not** wrong. We wrote “**such as**”, i.e. we included **examples** of factors that affect the assembly of **both or only one** of the subunits and referred to **original research papers** that include pulse chase experiments. To avoid confusion, we have modified the text to emphasize that **there is a long list of assembly factors that are required for assembly of only one of the subunits** (Klinge and Woolford, 2018; Woolford and Baserga, 2013). Ribosomal proteins are also only required for the assembly on one, but not the other, subunit. **That is, repression of the expression of the genes for subunit-specific factors or r-proteins will only affect production of one of the subunits.** This logically follows from the classification of these proteins as subunits-specific. **This is key to our approach: our aim is to understand what happens when you block assembly of one, but not the other, subunit.** We have also replaced the references to original papers with the reviews by Klinge and Woolford (2018) and Woolford and Baserga (2013).

3. Mutations in genes for r-proteins and subunit-specific factors can distort the normal 1:1 production of the two ribosomal subunits, thereby affecting the translation process. For example, an excess of 40S subunits may sequester mRNA in initiation complexes that cannot be converted to translating ribosome because of the shortage of 60S subunits.

Each of these two sentences need a reference.

We have rewritten the sentence to make clear that **because many factors and all ribosomal proteins are required for assembly of only one of the subunits** (see our response to point 2), **it logically follows** that abrogating the synthesis of these proteins only affect the assembly on one subunit and that the production of subunits in a the 1:1 ratio therefore must be disturbed. This is logic. No reference required. The point of our project was to investigate how cells respond to that, as is clearly stated in the Abstract and Introduction.

We have eliminated the last sentence to which the referee refers, since the topic is discussed later in the Discussion.

February 25, 2019

RE: Life Science Alliance Manuscript #LSA-2018-00150-TRR

Dr. Lasse Lindahl
University of Maryland, Baltimore County
Biological Sciences
1000 Hilltop Circle
Baltimore 21250

Dear Dr. Lindahl,

Thank you for submitting your Research Article entitled "The small and large ribosomal subunits depend on each other for stability and accumulation". I appreciate the introduced changes and think that they clarify the concerns that were raised by ref#3 and improve the manuscript. It is thus a pleasure to let you know that your manuscript is now accepted for publication in Life Science Alliance. Congratulations on this interesting work.

DISTRIBUTION OF MATERIALS:

Again, congratulations on a very nice paper. I hope you found the review process to be constructive and are pleased with how the manuscript was handled editorially. We look forward to future exciting submissions from your lab.

Sincerely,

Andrea Leibfried, PhD
Executive Editor
Life Science Alliance
Meyerohofstr. 1
69117 Heidelberg, Germany
t +49 6221 8891 502
e a.leibfried@life-science-alliance.org
www.life-science-alliance.org